

**Technical Note – AQMEII4 Activity 1: Evaluation of Wet and Dry Deposition Schemes as an Integral Part of**
**Regional-Scale Air Quality Models**
Stefano Galmarini[1][*], Paul Makar[2], Olivia E. Clifton[3], Christian Hogrefe[4][#], Jesse O. Bash[4], Roberto Bellasio [5], Roberto
Bianconi[5], Johannes Bieser[6], Tim Butler[7], Jason Ducker[8], Johannes Flemming[9], Alma Hodzic[3], Christopher D. Holmes[7],
Ioannis Kioutsioukis[10], Richard Kranenburg[11], Aurelia Lupascu[7], Juan Luis Perez-Camanyo[12] , Jonathan Pleim[4], Young-
Hee Ryu[13] Roberto San Jose[12], Donna Schwede[4], Sam Silva[14], Marta Garcia Vivanco[15], Ralf Wolke[16]
[1]Joint Research Center, European Commission, Ispra, Italy
[2]Air Quality Modelling and Integration Section, Environment and Climate Change Canada, Toronto, Canada
[3]National Center for Atmospheric Research, Boulder, CO, USA
[4]Office of Research and Development, U.S. Environmental Protection Agency, Research Triangle Park, NC, USA
[5]Enviroware srl, Concorezzo, MB, Italy
[6]Institute of Coastal Research, Helmholtz-Zentrum Geesthacht, Germany
[7]Institute for Advanced Sustainability Studies, Potsdam, Germany
[8]Earth, Ocean and Atmospheric Science, Florida State University, Tallahassee, FL, USA
[9]European Centre for Medium-Range Weather Forecasts, Reading, U.K.
[10] Laboratory of Atmospheric Physics, Department of Physics, University of Patras, Greece
[11]Netherlands Organization for Applied Scientific Research (TNO), Utrecht, The Netherlands
[12]Technical University of Madrid (UPM), Madrid, Spain
[13] Pohang University of Science and Technology (POSTECH), Pohang, South Korea
[14]Pacific Northwest National Laboratory, Richland, WA, USA
[15]CIEMAT, Madrid, Spain
[16]Leibniz Institute for Tropospheric Research, Leipzig, Germany
Corresponding authors: (*) Stefano.galmarini@ec.europa.eu, (#) Hogrefe.christian@epa.gov



**Abstract**
We present in this technical note the research protocol for Phase 4 of the Air Quality Model Evaluation International
Initiative (AQMEII4). This research initiative is divided in two activities, collectively having three goals: (i) to define
the current state of the science with respect to representations of wet and especially dry deposition in regional
models, (ii) to quantify the extent to which different dry deposition parameterizations influence retrospective air
pollutant concentration and flux predictions, and (iii) to identify, through the use of a common set of detailed
diagnostics, sensitivity simulations, model evaluation, and reducing input uncertainty, the specific causes for the
current range of these predictions. Activity 1 is dedicated to the diagnostic evaluation of wet and dry deposition
processes in regional air quality models (described in this paper), and Activity 2 to the evaluation of dry deposition
point models against ozone flux measurements at multiple towers with multiyear observations (in a subsequent
publication). The scope of these papers is to present the scientific protocols for AQMEII4, as well to summarize the
technical information associated with the different dry deposition approaches used by the participating research
groups of AQMEII4. In addition to describing all common aspects and data used for this multi-model evaluation
activity, most importantly, we present the strategy devised to allow a common process-level comparison of dry
deposition obtained from models using sometimes very different dry deposition schemes. The strategy is based on
adding detailed diagnostics to the algorithms used in the dry deposition modules of existing regional air quality
models, in particular archiving land use/land cover (LULC)-specific diagnostics and creating standardized LULC
categories to facilitate cross-comparison of LULC-specific dry deposition parameters and processes, as well as
archiving  effective conductance and effective flux as means for comparing the relative influence of different
pathways towards the net or total dry deposition. This new approach, along with an analysis of precipitation and
wet deposition fields, will provide an unprecedented process-oriented comparison of deposition in regional air-
quality models. Examples of how specific dry deposition schemes used in participating models have been reduced
to the common set of comparable diagnostics defined for AQMEII4 are also presented.



## 1. Introduction

Since 2009, the Air Quality Model Evaluation International Initiative (AQMEII, Rao et al., 2011) has focused on
evaluating regional-scale air quality models used for research and regulatory applications. The goal of AQMEII is to
conduct coordinated research projects and model inter-comparisons to advance model evaluation practices and
inform model development. This initiative is promoted by the European Commission Joint Research Center, the U.S.
Environmental Protection Agency (EPA) and Environment and Climate Change Canada and involves the regional-
scale air quality research communities active in both North America and Europe.
AQMEII has been executed in phases that each focused on a critical aspect of modelling systems. The phases were
conducted as multi-model comparisons that were analyzed through the organization of common modelling activities
and supported by gathering specific monitoring data needed to evaluate model performance. Each of the phases
required developing innovative evaluation and data reconciliation techniques to provide scientific insight across
disparate modeling systems. AQMEII phase 1 provided the first detailed annual ensemble comparison of air-quality
model predictions for North America and Europe (Galmarini et al., 2012). AQMEII phase 2 examined the impacts of
feedbacks between air-quality and weather on forecasting skill and identified the key sources of uncertainty in
feedback model forecasts (Galmarini et al., 2015). AQMEII phase 3, in collaboration with the Task Force on
Hemispheric Transport of Air Pollution (TF HTAP) (http://www.htap.org), studied the effects of intercontinental
transport on regional air quality predictions (Galmarini et al., 2017). Details and findings of the past three phases of
AQMEII can be found in journal special issues dedicated to these activities (Galmarini et al., 2012, 2015, 2017). The
AQMEII initiative is based on the four pillars of model evaluation described by Dennis et al. (2010): operational,
diagnostic, dynamic, and probabilistic evaluation, which will be partly described hereinafter.
This fourth phase of AQMEII (AQMEII4), detailed in this special issue and introduced by a pair of technical notes,
focuses on the processes of wet and especially dry deposition, including the parameterized approaches used within
current air quality models, and how these approaches and the details of their implementation influence model
predictions and performance across multiple modelling systems. Deposition is critical to the lifecycle of a pollutant,
as it regulates the rate of pollutant removal from the atmosphere and determines the net flux of that pollutant to
the earth's surface. This latter point is particularly important when the pollutants have a known deleterious effect
on ecosystems (e.g. the deposition of acidifying compounds to aquatic ecosystems, or the dry deposition of ozone
on vegetation). By affecting the pollution remaining in the atmosphere, deposition estimates also modulate
predictions of ambient pollutant concentrations that affect human health through inhalation exposure.
Deposition has only been peripherally investigated in past phases of AQMEII. The operational evaluation of air
quality models, in which modelled concentrations are directly compared to monitoring network observations,
quantifies the extent to which an air quality model meets expected performance. However, operational evaluation
does not provide the process-level understanding of the extent to which the performance results from correct
representation of model physical and chemical processes. In this context, dry and wet deposition are key processes
within air quality models because they represent removal, which can affect the concentrations of key atmospheric



species. Several past AQMEII publications were dedicated specifically to wet and dry deposition (Vivanco et al. 2018,
Hogrefe et al. 2020, Solazzo et al. 2018). However, only wet deposition fluxes could be evaluated against
observational data in these papers. The causes of differences in model predictions for dry deposition were not
determined. Some of the studies performed within AQMEII also addressed dynamic evaluation (i.e. the performance
of a model in capturing changes in concentrations or deposition fluxes when subjected to variations in meteorology
or emissions). The effects of these variations on deposition were therefore investigated, but without analysis at the
process level on the extent to which the details of deposition algorithms influenced model performance.
Recent studies of dry deposition of ozone have been fueled by the need to quantify impacts on global-to-regional
water and carbon cycles (Lombardozzi et al., 2015; Oliver et al., 2018), vegetation damage including crop yields
(McGrath et al., 2015; Emberson et al., 2018; Schiferl and Heald, 2018; Hong et al., 2020), and ozone air pollution
(Andersson and Engardt, 2010; Silva and Heald, 2018; Baublitz et al., 2020). In particular, reduced stomatal dry
deposition of ozone during droughts may contribute to high ozone pollution episodes (Vautard et al., 2005; Solberg
et al., 2008; Emberson et al., 2013; Huang et al., 2016; Anav et al., 2018; Lin et al., 2020).  Dry deposition of ozone
occurring through nonstomatal deposition pathways, on average 45% of the total (Clifton et al., 2020a), has also
been shown to be more variable and more important than predicted by current chemical transport models, with
implications for background and extreme ozone pollution (Clifton et al., 2017, 2020b). Previous intercomparisons at
the global scale suggest large differences in simulated ozone deposition velocities with implications for the simulated
tropospheric ozone budgets and the models' ability to quantitatively capture the drivers of recent trends and
interannual variability in observed ozone pollution (Hardacre et al., 2015; Wong et al., 2019).  However, process-
oriented evaluation in regional-to-global models is missing, in large part because key process-oriented diagnostics
have not been archived and different land use / land cover (LULC) inputs across models have inhibited the systematic
elucidation of processes driving the noted differences (Hardacre et al., 2015; Clifton et al., 2020a).  One way in which
discrepancies between observed and modelled deposition has been addressed is through model-measurement
fusion approaches (Schwede and Lear, 2014; Makar et al., 2018, Robichaud et al., 2019, Robichaud et al., 2020). Such
approaches could benefit from an improved characterization of process-level uncertainty in modeled dry deposition.
Despite the great advancements in regional-scale air quality modelling, the primary schemes used for dry and wet
deposition in today's models originated in the 1980's and 1990's. Moreover, while the role of deposition as a
persistent sink has been known for a long time (e.g. Chang et al., 1987; Irving and Smith, 1991; Borrell and Borrell,
2000), its relative importance in regulating trace species budgets has become more prominent in recent years as the
magnitude of the anthropogenic emission source term has generally decreased. The evaluation studies performed
within AQMEII (e.g., Solazzo et al. 2017; Hogrefe et al., 2018) and other recent work reaffirmed that deposition is a
process of paramount importance within an air quality model (e.g., Knote et al., 2015; Huang et al., 2016; Beddows
et al., 2017; Matichuk et al., 2017; Campbell et al., 2019; Sharma et al., 2020) with consequences of primary
relevance in a number of sectors (human health, agriculture, forestry, hydrology, soil management, ecosystems
management). Thus, there is renewed focus on better characterization of this term and its magnitude.



All the above points were the motivation to make use of the AQMEII community and evaluation infrastructure to
construct an AQMEII phase dedicated to deposition. This phase was designed to compare deposition predictions
from multiple regional models by isolating specific deposition pathways across multiple modelling systems and
across multiple LULC classification systems using common diagnostic tools. Analyzing dry deposition of gaseous
species, especially ozone and nitrogen species, is a particular focus, as is quantifying the range of model predictions
for acidifying wet and dry deposition. A process-level diagnostic intercomparison of particle dry deposition is not
conducted here due to the complexity added by model-to-model differences in the representation of aerosols (size
and composition) themselves. We also note that some previous work (e.g. Makar et al., 2018) suggests that the
impact of particle deposition on total nitrogen and sulphur deposition is relatively small, although particle deposition
is the main source of base cations transferred from the atmosphere to ecosystems. However, more recent work
(Saylor et al., 2019, Emerson et al., 2020) suggests that particle dry deposition algorithms used in current modelling
systems are highly uncertain, suggesting a need for performing further process-level diagnostic intercomparisons.
AQMEII4 has the following research goals:
• Quantify the performance and variability of dry and wet deposition fields simulated by multiple state-of-
the science regional air quality models.
• Document deposition schemes and key parameters used in these models in a framework that allows their
easy intercomparison.
• Identify and quantify the causes of differences in model-generated deposition fluxes by using detailed
ancillary diagnostic fields added to deposition algorithms and common LULC categories.
• Analyze dry deposition module performance with single-point model simulations driven by observation
data collected at towers with ozone flux measurements, and quantify the impacts of different conditions,
processes and parameters on simulated dry deposition (Activity 2; to be covered in a companion technical
note).
• Investigate methods for using simulated meteorological, concentration, and deposition fields from multiple
models in conjunction with available observations to estimate maps of total deposition and their
environmental impacts, including the prediction of exceedances of critical loads.

Most model dry deposition schemes are derived from Wesely (1989). However, their implementation in regional
and global models has considerable variation (a comparison with global models may be found in Hardacre et al.,
2015). Specifically, most schemes follow the parameterization structure used by Wesely (1989) but may differ in the
details of their representation of individual parameters and processes. This is discussed in more detail in Section 3.
In addition, dry deposition algorithms require, as a key input, information on LULC and vegetation. It is therefore
important to determine how the deposition modules themselves work, both as standalone physical descriptions,
and within a regional air quality model. AQMEII4 has been organized as two parallel activities to address the research
goals outlined above. AQMEII4 Activity 1 (introduced in this technical note) focuses on the detailed diagnostic





comparison of predictions of air quality model deposition fields, along with evaluation of model concentration and
wet deposition flux performance at routine monitoring stations in North America (NA) and Europe (EU). Activity 2
(introduced in a separate technical note) evaluates only the dry deposition schemes used in air quality models, and
other models used for impacts assessments, as zero-dimensional single-point models, driven by observed
meteorology, biophysics and ecosystem characteristics, at specific sites across the Northern Hemisphere where
ozone flux measurements have been collected continuously over at least a year, with many datasets spanning three
years or more. AQMEII4 will provide the most comprehensive analyses yet performed on dry deposition schemes,
since the schemes will be tested both within and independently from the air quality model, under controlled
conditions, and when subjected to variable meteorological and surface characteristic conditions. The single-point
modelling component allows a very detailed analysis of how ozone dry deposition is modeled; recent work
comparing 5 deposition algorithms at a single site (Wu et al., 2018) here has been extended to multiple sites,
additional deposition algorithms, and takes advantage of a new collection of ozone flux measurements at sites
around the Northern Hemisphere and new process-oriented diagnostics.
This technical note is the first of two which are designed to summarize all relevant information that constitute the
set up and organization of AQMEII4. The intent of these technical notes is to provide both the readers and authors
of this Special Issue with a common reference for the description of the AQMEII4 aims, scientific protocols, and
analysis approaches, the model reporting framework, the model input data and monitoring data used for model
evaluation, and descriptions of the model deposition algorithms themselves. By serving as common point of
reference for the individual studies undertaken through the AQMEII4 framework, these technical notes reduce the
need for repetition of background material by individual study papers which allows these papers to focus on specific
analyses and the presentation of the results of AQMEII4. They also allow the reader to access all relevant background
material in a single location rather than spread out over several papers. Because of this design, these technical notes
should not be viewed as stand-alone scientific papers as they do not contain any results, but rather as laying the
groundwork for subsequent scientific papers contributed by modeling groups to the AQMEII4 Special Issue. This first
technical note is dedicated to Activity 1 while the second is dedicated to Activity 2.

## 2. AQMEII4 Activity 1 Description

Activity 1 like the previous phases of AQMEII includes the evaluation of regional air quality model simulation on the
NA, EU, or both domains for at least a one-year period. Prior to describing the requested output that pertains strictly
to dry deposition, we briefly summarize in this section the modeling periods and domains, common inputs, and
standard concentration, meteorology, and wet deposition outputs for Activity 1.

### 2.1 Modeling Periods and Domains

For AQMEII4 Activity 1 the air quality community listed in Table 1 has been asked to perform two annual simulations
of the air quality over NA and/or EU.




| Group/Institution | Modeling System | Model Domains |
|---|---|---|
| Leibniz Institute for Tropospheric Research (TROPOS), Germany | COSMO / MUSCAT | EU |
| Environment and Climate Change Canada (ECCC), Canada | GEM / MACH (3 different model configurations) | NA |
| Technical University of Madrid (UPM), Spain | WRF-Chem | EU and NA |
| Netherlands Organization for Applied Scientific Research (TNO), The Netherlands | LOTOS / EUROS | EU |
| Institute for Advanced Sustainability Studies (IASS), Germany | WRF-Chem | EU and NA |
| US Environmental Protection Agency, USA | WRF / CMAQ (2 different model configurations) | NA |
| Helmholtz-Zentrum Geesthacht, Germany | COSMO-CLM / CMAQ | EU and NA |
| National Center for Atmospheric Research (NCAR), USA | WRF-Chem | NA |
| University of Hertfordshire, United Kingdom | WRF / CMAQ | EU |
| Research Centre for Energy, Environment and Technology (CIEMAT), Spain | ECMWF/IFS / CHIMERE | EU |


**Table 1. Participating institutes, models names and cases simulated**

Specifically, the years of interest in AQMEII4 are: North America - 2010 and 2016; Europe - 2009 and 2010. The NA
years were selected due their past use in policy-relevant emissions scenario simulation, with changes in emission
policies that may affect the deposition. In the case of Europe, the years illustrated a marked difference in
meteorological signatures between the two years, hence providing a gauge of the impact of meteorological
variability on deposition. Modeling multiple years also allows the investigation of the variability of impacts of
emission policies and weather conditions on deposition patterns.
All modeling groups carried out simulations on their own grid projections. These "native grid" simulations were
interpolated to a common 0.125° x 0.125° latitude-longitude grid over each continent to allow direct comparison of
gridded model data:
NA: 130°W <-> 59.5°W, 23.5°N <-> 58.5°N,
EU: 30 W <-> 60°E, 25°N <-> 70°N


Modeling groups are expected to perform their simulations on a grid with comparable-to-higher horizontal
resolution as these reported grids. The interpolation of model results from the native modeling grid to the common
analysis grid was recommended to use a mass conserving method for concentrations and fluxes and the nearest
neighbor method for diagnostic variables.
***2.2. Model Inputs Shared By All Participants***
Air-quality models require input fields for meteorology, emissions and chemical boundary conditions; differences in
each of these fields lead to differences in model results. All AQMEII exercises have considered the driving
meteorology to be an integral part of each participating model (for on-line models, such as studied under AQMEII-2
chemistry and meteorology are inseparable, since both are included in the same modelling platform) and have
therefore not attempted to harmonize meteorological fields across participants.  However, variations caused by
different emissions and chemical boundary conditions are removed in all AQMEII phases by requiring all participating
models to use a common set of emissions and lateral chemical boundary conditions (Galmarini et al., 2012, 2015,
2017). Note that due to their dependence on model-specific LULC and meteorology, biogenic emissions are not
prescribed and are generated by each group. For AQMEII4, the common model inputs were prepared as follows:
*2.2.1 Anthropogenic Emissions*
Emissions for anthropogenic sources over NA were prepared from U.S., Canadian, and Mexican inventory data using
the emissions processing approach developed for U.S. EPA "emission modeling platforms" (EMP). An EMP includes
not only the underlying point source, county or province level inventory data but also controls the temporal and
spatial allocation and chemical speciation of these inventories. For 2010, the processing was based on the "2011v6.3
EMP" (https://www.epa.gov/air-emissions-modeling/2011-version-63-platform). Year specific adjustments for 2010
were made to the EMP for several sectors (e.g. electric generating units, mobile sources, and residential wood
combustion) and Canadian emissions were based on a 2010 inventory rather than the 2013 inventory projected to
2011 used in the EMP. For 2016, the processing was based on the "2016beta EMP" (https://www.epa.gov/air-
emissions-modeling/2016v72-beta-and-regional-haze-platform)  which  is  documented  at
http://views.cira.colostate.edu/wiki/wiki/10197. These EMP were used by the US EPA to generate 8 different hourly
speciated files for each day in 2010 (1 gridded file with low-level emissions and files with elevated sources from 7
different sectors) and 9 different hourly speciated files for each day in 2016 (1 gridded file with low-level emissions
and files with elevated sources from 8 different sectors) which were then shared with all participants. Speciation
was performed for both the CB6R3 and SAPRC07 mechanism to provide flexibility to participants to map emissions
to the chemical mechanism used in their model. The same data were used by Environment and Climate Change
Canada to generate day-specific emissions for the GEM-MACH air-quality model, for the ADOMII mechanism used
within that model.  Annual gridded anthropogenic emissions using the Standard Nomenclature for Air Pollution
(SNAP) sector classification scheme were prepared over EU by TNO for 2009 and 2010 as part of the MACC-III project
(Kuenen et al., 2015) and were provided to EU modeling groups along with reference temporal allocation and
speciation profiles. If necessary, EU modeling groups used other emission datasets available to them to fill in





emissions near the edges of their modeling domains if their modeling domains extended beyond the are covered by
the MACC-III emissions provided by TNO.

*2.2.2 Forest Fire Emissions*

The forest fire emissions over NA for 2010 were a combination of emissions over the U.S. included in the "2011v6.3"
EMP and emissions over Canada provided by Environment and Climate Change Canada (ECCC) while 2016 forest fire
emissions over both the U.S. and Canada were obtained from the "2016 beta" EMP. Data distributed to modeling
groups included both the mass of emissions of Criteria Air Contaminants (speciated into the gases of the gas-phase
chemistry mechanisms noted above) and the parameters necessary to compute plume rise using a prescribed plume
rise algorithm based on the large stack plume rise formula of Briggs (Briggs, 1971, 1972).  While different modelling
platforms often have their own approaches for estimating forest fire emissions, particularly in an operational
context, as was the case for anthropogenic emissions, this unified approach was adopted in order to reduce the
variability in model performance associated with emissions inputs. Forest fire emissions for 2009 and 2010 over EU
were provided by the Finnish Meteorological Institute and were developed using the IS4FIRESv2 methodology
described in Soares et al. (2015). These emissions were vertically allocated to eight layers with heights ranging from
50m to 6200m, with individual groups re-allocating the resulting mass to their own vertical discretization.

*2.2.3 NO emissions from lightning*

Although previous phases of AQMEII did not consider NO emissions from lightning, these emissions were included
in the current phase due to their impact on nitrogen deposition fluxes. To provide a unified forcing from this source
across all models, the emissions were based on the GEIA monthly climatology (Price et al., 1997) rather than in-line
parameterizations based on meteorological fields implemented in some but not all participating models. Although
using climatological lightning does not capture the linkage between modeled meteorology and NO emission from
lightning, this approach ensures that the bulk effects are included in all modeling systems and streamlines the
interpretation of the modeling results by removing a potential difference in emissions input. The monthly
climatological values were allocated diurnally based on Table 2 in Blakeslee et al. (2014) and distributed to
participating groups as 2-dimensional files. Groups were then asked to allocate these emissions to their specific
vertical grid based on Table 2 of Ott et al. (2010), using the tropical profiles for land and water (or an average of the
two) for grid cells with latitudes below 23.5N, the subtropical profile for grid cells with latitudes between 23.5°N and
40°N, and the midlatitude profile for grid cells with latitudes > 40°N.

*2.2.4 Chemical boundary conditions*

Concentrations of the 33 longer-lived trace gas and aerosol species listed in Table 2 were provided by the European
Centre for Medium-Range Weather Forecasts (ECMWF) for the two continents and for the modeled time periods so
that participants could prepare initial and boundary conditions for their regional-scale modeling domains. The
concentration fields were based on the Copernicus Atmospheric Monitoring Service (CAMS) EAC4 reanalysis product
(Inness et al., 2019) and were provided every 3 hours on a 0.75° x 0.75° grid with 54 vertical levels from the surface
to 2 hPa. The vertical grid structure varied in both resolution and vertical extent across models and individual





participants were responsible for interpolating the CAMS fields to their horizontal and vertical grid structure. The
CAMS species were matched by participants to their own internal model speciation (and, in the case of the
particulate matter emissions, to the particle size distribution of their own models).

| Trace Gas Species | Aerosol Species |
|---|---|
| $O_3$ (ozone) | Sea Salt Aerosol @80% relative humidity (wet radii 0.03 - 0.5 μm)[*] |
| CO (carbon monoxide) | Sea Salt Aerosol @80% relative humidity (wet radii 0.5 - 5 μm)[*] |
| NO (nitrogen monoxide; nitric oxide) | Sea Salt Aerosol @80% relative humidity (wet radii 5 - 20 μm)[*] |
| $NO_2$ (nitrogen dioxide) | Dust Aerosol @0% relative humidity (dry radii 0.03 - 0.55 μm) |
| PAN (peroxyacetyl nitrate) | Dust Aerosol @0% relative humidity (dry radii 0.55 - 0.9 μm) |
| $HNO_3$ (nitric acid) | Dust Aerosol @0% relative humidity (dry radii 0.9 - 20 μm) |
| $CH_2O$ (formaldehyde) | Hydrophobic Organic Matter Aerosol @0% relative humidity |
| SO2 (sulfur dioxide) | Hydrophilic Organic Matter Aerosol @0% relative humidity |
| $H_2O_2$ (hydrogen peroxide) | Hydrophobic Black Carbon Aerosol @0% relative humidity |
| $CH_3COCH_3$ (acetone) | Hydrophilic Black Carbon Aerosol @0% relative humidity |
| $C_2H_6$ (ethane) | Sulphate Aerosol @0% relative humidity |
| PAR (paraffins) | |
| $CH_3OH$ (methanol) | |
| $C_3H_8$ (propane) | |
| $C_2H_5OH$ (ethanol) | |
| $C_2H_4$ (ethene) | |
| ALD2 (aldehydes) | |
| OLE (olefins) | |
| $C_5H_8$ (isoprene) | |
| HCOOH (formic acid) | |
| $CH_3OOH$ (methylperoxide) | |
| ONIT (organic nitrates) | |
| [*]based on guidance from ECMWF, participants were advised to transform the provided values back to dry matter by applying a reduction factor of 4.3 for the mass mixing ratios and a reduction factor of 1.99 for the radii of the sea salt bin limits | |


**Table 2. Variables from the CAMS EAC4 reanalysis provided for the generation of initial and boundary conditions.**
*2.3 Standard Model Outputs*
We distinguish here between model output similar in scope and intent to previous ensemble model comparisons in
past phases of AQMEII (i.e., "standard model outputs"), and the detailed diagnostic outputs reported under
AQMEII4. The standard output requested from all participating models comes in two major forms: as hourly gridded





surface concentrations and meteorological variables on the common grids described earlier, and as model values
extracted at monitoring network station locations. Tables A1 – A3 of Appendix A list the variables requested for gas
and particle phase species, meteorology, and grid scale deposition fluxes. The meteorological variables have been
extended considerably compared to past phases of AQMEII, to include more parameters that describe the planetary
boundary layer. The gridded fields of integrated emissions were also requested as output, to be used to check that
the right amounts of masses were inputted into the models.
A list of all available surface monitoring locations in both continents for concentrations of gas- and particle-phase
species, precipitation chemistry, and meteorology was distributed to the AQMEII4 participants who are expected to
produce model results for all species presented in Appendix A for the grid location closest to the monitor or
interpolated to the monitoring. In particular, we note that the analysis of wet deposition in AQMEII4 will rely on the
precipitation and wet deposition flux variables listed in Table A3.  In addition, the locations where vertical profiles
of ozone are routinely measured in NA and EU are also provided and modelling groups were expected to produce
the ozone vertical profiles at those spots. For more information on the routine monitoring networks used in AQMEII
please refer to Galmarini et al. (2012, 2015, 2017).

**3. Strategy For The Diagnostic Intercomparison Of Dry Deposition From Different Grid-Based Models**
Analysis of dry deposition is the focus of AQMEII4. In particular, AQMEII4 intends to go beyond an operational
evaluation of ambient concentrations and comparison of total deposition across models because this approach does
not provide enough information to determine the causes of different deposition totals among regional models. The
novelty of AQMEII4 is that we request additional and very detailed diagnostic-evaluation outputs related to dry
depositional from all of the models. With these very detailed outputs, we can compare the important elements of
the model machinery and understand model differences.
Many regional models use the Wesely (1989) dry deposition scheme, but several variants have been developed and
implemented with different levels of sophistication. Dry deposition schemes are mostly resistance frameworks – by
framework, we mean the structure of the scheme with respect to how processes relate to one another – and all of
the regional models in AQMEII4 use resistance frameworks for dry deposition. Resistance frameworks are based on
the representation of series and parallel resistors in electrical circuits. Differences in resistance frameworks across
regional models imply that comparing a given process among the regional models is not straightforward. Thus,
diagnostic variables that account for differences in resistance frameworks need to be reported. Below, we present
the strategy devised to reduce any dry deposition scheme to the essential set of comparable variables regardless of
the differences in the frameworks of the schemes that generated them.

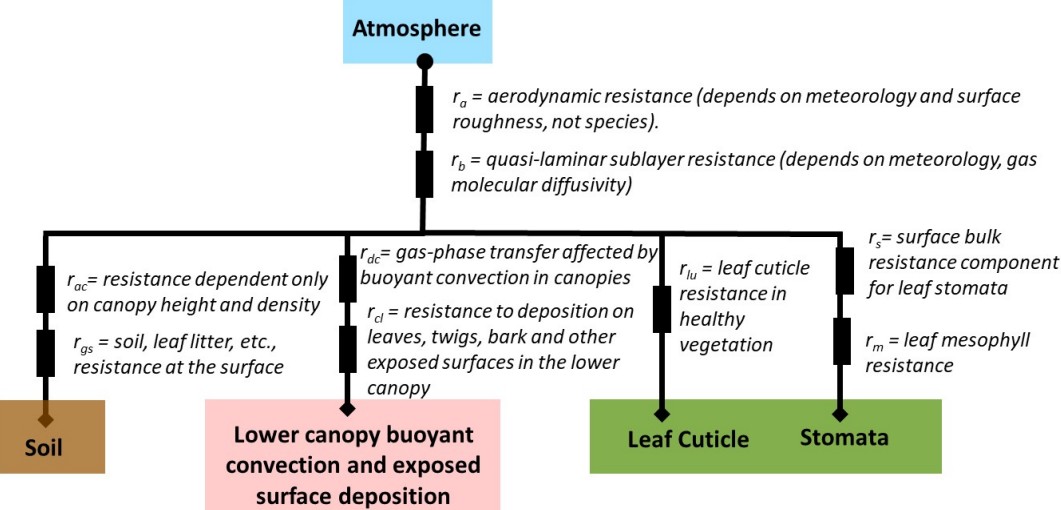

**Figure 1. Schematic of the resistance framework for gas-phase dry deposition for the Wesely (1989) scheme. Circles and diamonds show where ozone concentration is needed as input for a given framework. At the diamonds, the ozone concentration is assumed to be zero. Rectangles indicate resistances.**

We start with a description of the Wesely (1989) resistance framework, one of the earliest literature examples of a resistance framework for dry deposition and arguably the most popular dry deposition scheme, and follow with both generic and specific examples of other resistance frameworks as a guide to the AQMEII4 output protocol. The components of the deposition velocity are process-based resistances (units s cm$^{-1}$) that impede the transfer of mass to a variety of surfaces. Resistances are added in series for processes operating on the same depositional pathway, and in parallel when multiple surfaces for dry deposition exist. In the original Wesely (1989) scheme, four deposition pathways were used: soil, "lower canopy and exposed surfaces", leaf cuticles, and plant stomata. Gases are first impeded by an aerodynamic resistance to deposition ($r_a$), second impeded by a quasi-laminar sublayer resistance ($r_b$), and third impeded by a bulk surface resistance term ($r_c$) composed of a parallel summation of the resistances associated with each pathway. The three impedances to deposition are added into a total resistance, the inverse of which is the deposition velocity of the gas (units cm s$^{-1}$) :

$$v_d = (r_a + r_b + r_c)^{-1} \qquad (1)$$

The bulk surface resistance ($r_c$) in Wesely (1989) follows:

$$r_c = \left( (r_s + r_m)^{-1} + (r_{lu})^{-1} + (r_{dc} + r_{cl})^{-1} + (r_{ac} + r_{gs})^{-1} \right)^{-1} \qquad (2)$$

The component resistances used in r$_c$ are defined in Figure 1, which is a schematic of the Wesely (1989) resistance framework.



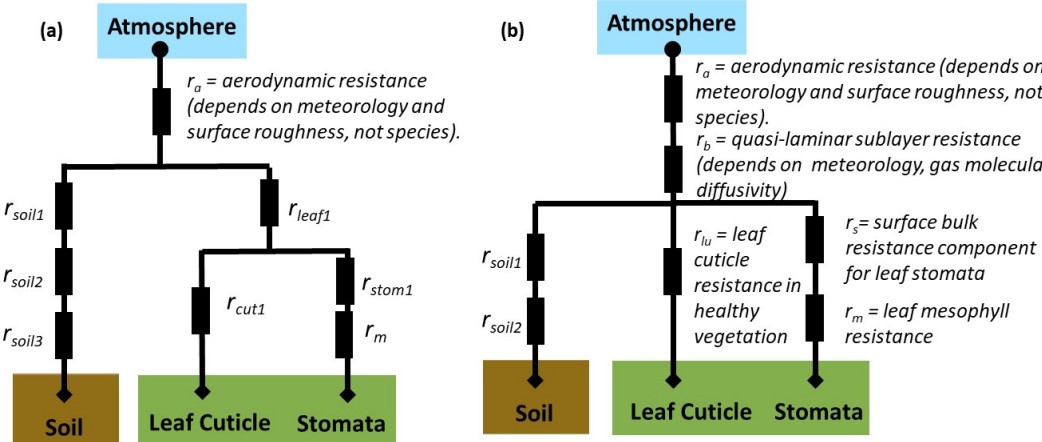


**Figure 2. Two generic deposition resistance examples.**
Work subsequent to Wesely (1989) also uses the resistance approach, but sometimes with considerable variation in
the resistance framework, the number of surfaces to which dry deposition occurs, and/or the processes represented
by individual resistances. Schematics of resistance frameworks as two generic examples are shown in Figure 2. In
these examples, the Wesely (1989) deposition pathway for "lower canopy buoyancy and exposed surfaces"
deposition is not included. The example of Figure 2(a) also lacks a quasi-laminar sublayer resistance $r_b$ applied across
all surface types. Instead, surface-specific quasi-laminar sublayer resistances are used: $r_{soil2}$ for soil and $r_{leaf1}$ for
leaves. The examples in Figure 2 demonstrate two ways in which the resistance framework has been adapted from
Wesely (1989). In general, the diversity in resistance frameworks across models complicates model intercomparison
of individual resistances.

When there are differences in resistance frameworks across models, the deposition pathways may be compared
across models using a construct we will refer to here as *effective conductance* (Paulot et al., 2018; Clifton et al.,
2020b). While generally a conductance is simply the inverse of a resistance, an *effective* conductance is the
contribution of a given depositional pathway to the deposition velocity, expressed in the same units as the
deposition velocity. The sum of the effective conductances for all deposition pathways is the deposition velocity.
The effective conductances of the soil ($E_{SOIL}$), lower canopy ($E_{LCAN}$), cuticle ($E_{CUT}$) and stomata ($E_{STOM}$) branches
specifically for Wesely (1989) are given by[1]:
$$E_{SOIL} = \left( \frac{(r_{ac}+r_{gs})^{-1}}{(r_s+r_m)^{-1}+(r_{lu})^{-1}+(r_{dc}+r_{cl})^{-1}+(r_{ac}+r_{gs})^{-1}} \right) v_d \qquad (3)$$

---

[1] Note that the depositing gases in each pathway are influenced by $r_a$ and $r_b$ prior to encountering the different resistances that make up $r_c$, and this is why $v_d$, which includes the influence of $r_a$ and $r_b$, is scaled by the fraction of the inverse of $r_c$ occurring through a given pathway. Some models include surface-specific quasi-laminar sublayer resistances; when this is the case, these terms appear in the pathway-specific fractions of the total uptake terms.


$$E_{LCAN} = \left( \frac{(r_{dc}+r_{cl})^{-1}}{(r_s+r_m)^{-1}+(r_{lu})^{-1}+(r_{dc}+r_{cl})^{-1}+(r_{ac}+r_{gs})^{-1}} \right) v_d \tag{4}$$

$$E_{CUT} = \left( \frac{(r_{lu})^{-1}}{(r_s+r_m)^{-1}+(r_{lu})^{-1}+(r_{dc}+r_{cl})^{-1}+(r_{ac}+r_{gs})^{-1}} \right) v_d \tag{5}$$

$$E_{STOM} = \left( \frac{(r_s+r_m)^{-1}}{(r_s+r_m)^{-1}+(r_{lu})^{-1}+(r_{dc}+r_{cl})^{-1}+(r_{ac}+r_{gs})^{-1}} \right) v_d \tag{6}$$

The denominator in each of equations (3) to (6) is the inverse of the bulk surface resistance $r_c$ and the numerators are the inverses of the resistances associated with each pathway in $r_c$. We emphasize that the calculation of the effective conductances depends on the resistance framework used; equations (3) to (6) are specific to Wesely (1989) and require modification for other resistance frameworks, and we provide examples of formulae for these terms for other frameworks, in Section 4.1, and Appendix B. Calculation of the effective conductances requires either archiving all component resistances in a given framework and subsequent post-processing, or their online calculation.

For any given model, effective conductances are an invaluable tool for determining the extent to which each pathway impacts dry deposition velocity, and which deposition pathways drive spatiotemporal variability in dry deposition velocity. Key for AQMEII4, the effective conductances allow a cross-comparison of the main deposition pathways across different resistance frameworks. The primary terms of comparison for dry deposition schemes in AQMEII4 are thus the effective conductances. In addition, given that many models' resistance frameworks follow Wesely (1989), we also request those individual resistance terms held in common by most models, to allow exact comparisons of individual processes which may influence or control a given pathway. These resistances include:

(1) A term for the aerodynamic resistance, $r_a$.

(2) A term for the bulk resistance to deposition associated with surfaces $r_c$.

(3) A term or series addition set of terms describing the stomatal resistance, $r_s$.

(4) A term or series addition set of terms describing the mesophyll resistance $r_m$.

(5) A term or series addition set of terms describing the cuticle resistance, $r_c$.

(6) Terms to describe quasi-laminar sublayer resistance, $r_b$.

(7) A term to describe within-canopy buoyant convection, $r_{dc}$.

With regards to (6), the implementation of quasi-laminar sublayer resistance ($r_b$ in Wesely (1989)) tends to differ among models. Some models use the Wesely (1989) concept of a pathway-independent quasi-laminar sublayer resistance. Others use quasi-laminar sublayer resistances as pathway-dependent (e.g. Fig. 2a, where the $r_{soil2}$ and $r_{leaf1}$ represent quasi-laminar sublayer resistances for soil and leaf pathways, respectively). The quasi-laminar sublayer resistance is thus reported in AQMEII4 for each pathway, with the models for which the term is independent of pathway reporting the same value for each pathway. Pathway-dependent quasi-laminar sublayer resistances are to be reported as "not present" only if the given pathway does not exist in the framework.





Note that models that include a single deposition pathway to soil that incorporates $r_{dc}$ are requested to report that
pathway as "lower canopy" not "soil". For example, the LOTOS-EUROS dry deposition scheme (Fig. B4) reports the
effective conductance calculated for the soil pathway as $E_{LCAN}$ due to the presence of the in-canopy resistance term
in this pathway. In contrast, the CMAQ-M3DRY and CMAQ-STAGE dry deposition schemes (Figs. B2 and B3) have two
separate pathways for deposition to soil, one for vegetation-covered soil and one for bare soil.  Due to the inclusion
of the in-canopy convective resistance in the computations for vegetation-covered soil, the effective conductance
for that pathway is reported as $E_{LCAN}$, while the effective conductance for the bare soil pathway should be reported
as $E_{SOIL}$.
Specific resistance terms for the soil deposition pathway and the lower canopy pathway have not been requested
because the resistance frameworks for these pathways vary considerably across models and therefore specific
resistance terms are not easily comparable. For example, Wesely (1989) used a single term for the soil resistance
(Fig. 1) while other models may use two or three resistances related to dry deposition to soil only and added in series
(Fig. 2).
In addition to the effective conductances, another set of diagnostic fields is calculated during post processing: the
time-aggregated fractional mass (or charge equivalent) *flux* transferred to the surface via each of the four deposition
pathways (hereinafter, *effective flux*). The effective flux is calculated on an hourly basis prior to conversion to
AQMEII4 time-aggregated gridded and station data using ENFORM, and is the product of the hourly effective
conductances, dry deposition mass fluxes, and inverses of the deposition velocity. Effective *conductances* provide
an estimate of the importance of each pathway towards the deposition velocity. However, since the flux depends
on the deposition velocity and the near-surface air concentration, which both vary on hourly timescales, estimating
the aggregate importance of each deposition pathway towards the flux requires calculating the effective flux before
time-aggregation.
Figure 3 provides an example of the different yet complementary information resulting from effective conductances
and effective fluxes, showing hourly $SO_2$ concentrations, effective conductances, and effective fluxes for a boreal
forest impacted by a large industrial $SO_2$ stack sources, and hourly $NO_2$ concentrations, effective conductances, and
effective fluxes for a location to the north-east of New York City. In both cases, high concentrations of the pollutant
gas (Fig. 3a,d) occur at night, while the deposition velocity, due to the stomatal pathway (Fig. 3b,e), maximizes during
the day. As a result of the low daytime concentrations, the effective fluxes for $SO_2$ (Fig. 3c) show a relatively minor
contribution of the stomatal pathway to the deposited mass despite the major contribution of the stomatal pathway
to the daytime deposition velocity. As the result of high night and morning concentrations, the effective fluxes for
$NO_2$ (Fig. 3f) show separate day and night peaks of about equal magnitude, with the stomatal pathway dominating
daytime values, and roughly equivalent contributions from stomatal and soil pathways at night.



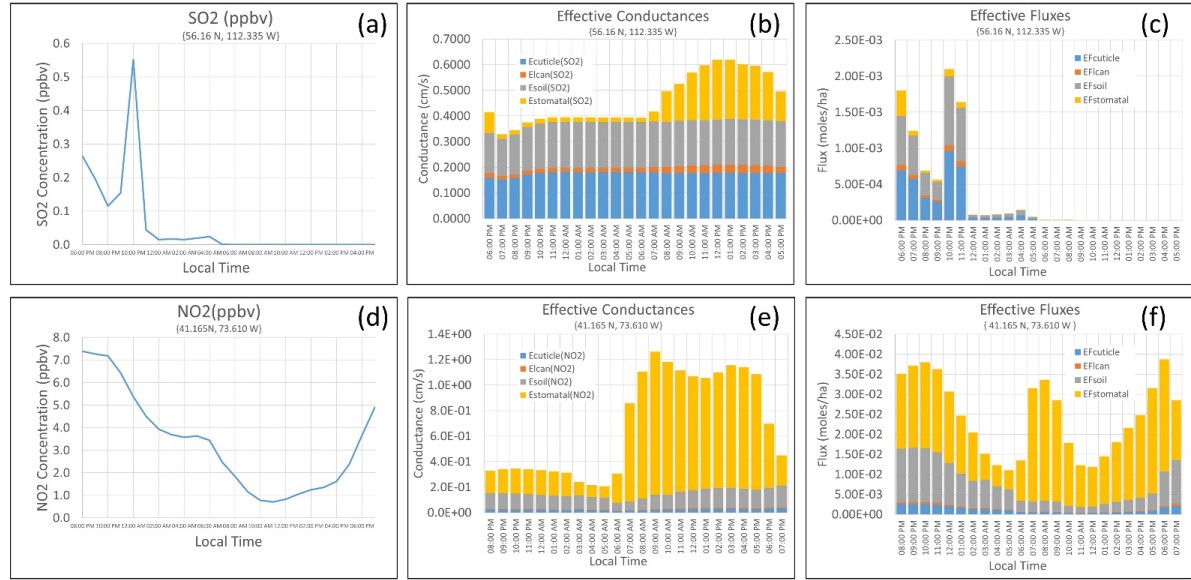


**Figure 3. Two examples of diurnal variations in concentrations (a, d), effective conductances (b, e), and effective fluxes (c, f) for SO₂ (top row) and NO₂ (bottom row).**

We also consider that dry deposition strongly depends on LULC type, and different models use unique LULC databases. We thus request LULC-specific variables along with the fractional areal coverage for each LULC type, which allows quantifying not only the impacts of different LULC specific processes and parameters on dry deposition, but also the impacts of different LULC databases. 'Generic' AQMEII4 LULC types were devised due to the use of a wide variety of LULC databases across air quality models, both in terms of the source of the data and the number of LULC types employed. The AQMEII4 LULC types listed in Table 2 are broad LULC types into which the model-specific LULC types could be aggregated, to allow intercomparison between models. Study participants aggregated their LULC-model-specific diagnostic outputs to the set of common AQMEII4 LULC types using the fractional representation of each native LULC type contributing to the AQMEII4 type within each grid cell. Generic AQMEII4 LULC types were constructed after analysis of the LULC schemes in the participating models. A suggested mapping between model and AQMEII4 LULC types was provided to participants, along with the instruction that the mapping actually employed should be reported. The grid cell fractions of both the native model LULC types, as well as the resulting fractions of AQMEII4 LULC types, were reported by participants. Note that there is a large variety in number and therefore types of LULC across models, and thus the each of the generic types represents a rather broad range of LULCs.

For AQMEII4, the terms listed in Table 4 were reported for SO₂, NO₂, NO, HNO₃, NH₃, PAN, HNO₄, N₂O₅, organic nitrates, O₃, H₂O₂ and HCHO, both as a function of the 16 generic AQMEII4 LULC types (Table 3) as well as for the net grid-scale calculation for each grid cell and/or receptor. Models employing bidirectional flux algorithms for the dry deposition of atmospheric NH₃ reported a different set of terms, given in Section 4.2.






| Generic LULC Categories for Remapping |
|---|
| Water |
| Developed / Urban |
| Barren |
| Evergreen needleleaf forest |
| Deciduous needleleaf forest |
| Evergreen broadleaf forest |
| Deciduous broadleaf forest |
| Mixed forest |
| Shrubland |
| Herbaceous |
| Planted/Cultivated |
| Grassland |
| Savanna |
| Wetlands |
| Tundra |
| Snow and Ice |


**Table 3 Generic land use / land cover types for AQMEII4**
Table 4 summarizes the diagnostic variables related to gaseous dry deposition reported by all participants, the
variable names as described in the AQMEII4 TSDs, and a description of each variable. Equations (2) through (6) and
the related text describe the terms specifically for the resistance framework of Wesely (1989); additional examples
for participating models' resistance frameworks are provided in the Appendix tables and figures.
The presence of surface wetness or snow is incorporated into the effective conductance, effective flux, and
component resistances. In other words, separate component resistances or effective conductances and fluxes for
snow-covered or wet surfaces were not reported. In order to compare the impacts of the different models'
predictions regarding snow cover or wetness, additional diagnostic variables were requested to describe surface
state (e.g. fractional snow cover and either the values of binary wet/dry conditions or fractions in surface wetness).

| Name | AQMEII4 Name | Formula |
|---|---|---|
| $V_d$ | VD | Deposition velocity |
| $r_a$ | RES-AERO | Aerodynamic resistance |
| $r_c$ | RES-SURF | Bulk surface resistance |



| $r_s$ | RES-STOM | Stomatal resistance |
|---|---|---|
| $r_m$ | RES-MESO | Mesophyll resistance |
| $r_c$ | RES-CUT | Cuticle resistance |
| $E_{STOM}$ | ECOND-ST | Effective conductance associated with deposition to plant stomata |
| $E_{CUT}$ | ECOND-CUT | Effective conductance associated with deposition to leaf cuticles |
| $E_{SOIL}$ | ECOND-SOIL | Effective conductance associated with deposition to soil and un-vegetated surfaces |
| $E_{LCAN}$ | ECOND-LCAN | Effective conductance associated with deposition to the lower canopy |
| $r_{b,stom}$ | RES-QLST | Quasi-laminar sublayer resistance associated with stomatal pathway* |
| $r_{b,cut}$ | RES-QLCT | Quasi-laminar sublayer resistance associated with cuticular pathway* |
| $r_{b,soil}$ | RES-QLSL | Quasi-laminar sublayer resistance associated with soil pathway* |
| $r_{b,lcan}$ | RES-QLLC | Quasi-laminar sublayer resistance associated with lower canopy pathway* |
| $r_{dc}$ | RES-CONV | Resistance associated with within-canopy buoyant convection |
| **Post Processing Fields:  Effective Conductances x Net flux / Deposition Velocity** | | |
| DFLX-LCAN | | Fraction of flux via lower canopy pathway |
| DFLX-ST | | Fraction of flux via stomatal pathway |
| DFLX-CUT | | Fraction of flux via cuticle pathway |
| DFLX-SOIL | | Fraction of flux via soil pathway |

**\*** = $r_b$ if this is pathway-independent for the resistance framework
**Table 4.  AQMEII4 reported dry deposition diagnostic variables for gas phase species.**
Gridded dry deposition diagnostic variables were archived as hourly values for the native LULC types, and then
converted to the generic AQMEII4 LULC types during post-processing. The ENFORM Fortran code provided to all
participants was used to convert gridded fields from the hourly values to temporal aggregations of the hourly values.
Hourly diagnostics were converted to "monthly median diurnal" values using ENFORM by taking the medians of all
values for a given UTC hour in a given month, thus reducing 8,760 hourly values for each year to 288 values (24 hours
x 12 months). The use of monthly median diurnal values is motivated by the need to reduce the amount of data to
be transferred and analyzed on a single server (despite this aggregation, each year of gridded model output requires
up to 200 Gb of storage), while preserving the key aspects of diurnal and seasonal variations.
The use of a median rather than an arithmetic mean for AQMEII4 diagnostic time aggregation resulted from
consideration of the manner in which different dry deposition algorithms deal with pathways that effectively shut
down under certain conditions. For example, some algorithms employ an upper-limit resistance to represent
conditions under which the pathway transmits little mass to the surface (e.g. nighttime stomatal resistances may be
set to very large values). Others simply use code branching to prevent a pathway from contributing to $r_c$ (e.g. the
entire stomatal pathway is removed from $r_c$ at night). Others employ different resistance frameworks for different
conditions (e.g. to account for snow-covered surfaces). However, the AQMEII4 protocol requires participants to





submit "missing values" as a specific code (-9) in order to allow filtering of valid from invalid data during time
aggregation. An algorithm removing a pathway may thus have a different number of valid values from an algorithm
employing a large resistance. Similarly, a seasonal transition where the resistance network changes depending on
whether a surface is snow-covered becomes difficult to interpret in an time-average, whereas time-median valid
values allow for a more meaningful comparison.
For example, if only 20% of the resistances at 14:00 LT in a given month and grid cell are snow covered, then the
monthly median for 14:00 LT would represent values typical of snow-free conditions, both for models representing
resistances under snow-covered conditions as missing, and models representing them as large values. Thus, the
monthly median comparison represents the most common conditions encountered during the month for both
models. On the other hand, while the monthly average resistance for 14:00 LT represents snow-free conditions for
the model that treats snow-covered hours as missing, the monthly average for the model that represents snow-
covered conditions as a large value is not meaningful and complicates inter-model comparison.
Monthly median diurnal values capture both seasonal and diurnal variations in the archived fields and allow
comparisons between algorithms shutting off a pathway by removing the pathway and algorithms shutting off a
pathway with high resistance values. Note that the same data completeness criterion used for comparing simulated
and observed ambient concentrations was employed here for the construction of the median values. Specifically,
more than 75% of the values within a month were required for a median to be constructed.
**4. More Example Calculations of AQMEII4 Dry Deposition Variables.**
**4.1 Variations to the Wesely (1989) Resistance Framework**
For the sake of clarity, we provide examples of how specific dry deposition schemes can be reduced to the common
set of variables described above. The generic schemes presented in Fig. 2a,b along with the Nemitz et al. (2001)
bidirectional scheme for $NH_3$ have been selected as examples here, while Appendix B provides additional examples
for specific schemes implemented in participating models. The AQMEII4 protocol and these specific examples
provide a standard form of representing key aspects of dry deposition schemes, which may be adopted by similar
activities or initiatives in the future.  Note that some of these example algorithms do not have a separate resistance
for lower canopy buoyant convection or a deposition pathway to the lower canopy and exposed surfaces, hence the
associated effective conductance (ECOND-LCAN) and resistance (RES-CONV and RES-QLLC) terms are not reported.

| Name | AQMEII4 Name | Formula |
|---|---|---|
| $r_a$ | RES-AERO | $RES\text{-}AERO = r_a$ |
| $r_c$ | RES-SURF | $RES\text{-}SURF = \left( (r_{leaf1} + ((r_{stom1} + r_m)^{-1} + (r_{cut1})^{-1})^{-1})^{-1} + (r_{soil1} + r_{soil2} + r_{soil3})^{-1} \right)^{-1}$ |
| $r_s$ | RES-STOM | $RES\text{-}STOM = r_{stom1}$ |





| | | |
|---|---|---|
| $r_m$ | RES-MESO | $RES\text{-}MESO = r_m$ |
| $r_c$ | RES-CUT | $RES\text{-}CUT = r_{cut1}$ |
| $E_{STOM}$ | ECOND-ST | $ECOND\text{-}ST =$ $\left(\frac{(r_{stom1}+r_m)^{-1}}{(r_{stom1}+r_m)^{-1}+(r_{cut1})^{-1}}\right)\left(\frac{(r_{leaf1}+((r_{stom1}+r_m)^{-1}+(r_{cut1})^{-1})^{-1})^{-1}}{(r_{leaf1}+((r_{stom1}+r_m)^{-1}+(r_{cut1})^{-1})^{-1})^{-1}+(r_{soil1}+r_{soil2}+r_{soil3})^{-1}}\right)V_d$ |
| $E_{CUT}$ | ECOND-CUT | $ECOND\text{-}CUT =$ $\left(\frac{(r_{cut1})^{-1}}{(r_{stom1}+r_m)^{-1}+(r_{cut1})^{-1}}\right)\left(\frac{(r_{leaf1}+((r_{stom1}+r_m)^{-1}+(r_{cut1})^{-1})^{-1})^{-1}}{(r_{leaf1}+((r_{stom1}+r_m)^{-1}+(r_{cut1})^{-1})^{-1})^{-1}+(r_{soil1}+r_{soil2}+r_{soil3})^{-1}}\right)V_d$ |
| $E_{SOIL}$ | ECOND-SOIL | $ECOND\text{-}SOIL = \left(\frac{(r_{soil1}+r_{soil2}+r_{soil3})^{-1}}{(r_{leaf1}+((r_{stom1}+r_m)^{-1}+(r_{cut1})^{-1})^{-1})^{-1}+(r_{soil1}+r_{soil2}+r_{soil3})^{-1}}\right)V_d$ |
| $E_{LCAN}$ | ECOND-LCAN | $ECOND\text{-}LCAN = -9$ |
| $r_{b,stom}$ | RES-QLST | $RES\text{-}QLST = r_{leaf1}$ |
| $r_{b,cut}$ | RES-QLCT | $RES\text{-}QLCT = r_{leaf1}$ |
| $r_{b,soil}$ | RES-QLSL | $RES\text{-}QLCL = r_{soil2}$ |
| $r_{b,lcan}$ | RES-QLLC | $RES\text{-}QLLC = -9$ |
| $r_{dc}$ | RES-CONV | $RES\text{-}CONV = -9$ |

**Table 5. AQMEII4 dry deposition diagnostic variables for gas phase species corresponding to the resistance**
**framework of Fig. 2a.**




| Name | AQMEII4 Name | Formula |
|---|---|---|
| $r_a$ | RES-AERO | $RES\text{-}AERO = r_a$ |
| $r_c$ | RES-SURF | $RES\text{-}SURF = ((r_s + r_m)^{-1} + (r_{lu})^{-1} + (r_{soil1} + r_{soil2})^{-1})^{-1}$ |
| $r_s$ | RES-STOM | $RES\text{-}STOM = r_s$ |
| $r_m$ | RES-MESO | $RES\text{-}MESO = r_m$ |
| $r_c$ | RES-CUT | $RES\text{-}CUT = r_{lu}$ |
| $E_{STOM}$ | ECOND-ST | $ECOND\text{-}ST = \left(\frac{(r_s+r_m)^{-1}}{(r_s+r_m)^{-1}+(r_{lu})^{-1}+(r_{soil1}+r_{soil2})^{-1}}\right)V_d$ |
| $E_{CUT}$ | ECOND-CUT | $ECOND\text{-}CUT = \left(\frac{(r_{lu})^{-1}}{(r_s+r_m)^{-1}+(r_{lu})^{-1}+(r_{soil1}+r_{soil2})^{-1}}\right)V_d$ |
| $E_{SOIL}$ | ECOND-SOIL | $ECOND\text{-}SOIL = \left(\frac{(r_{soil1}+r_{soil2})^{-1}}{(r_s+r_m)^{-1}+(r_{lu})^{-1}+(r_{soil1}+r_{soil2})^{-1}}\right)V_d$ |
| $E_{LCAN}$ | ECOND-LCAN | $ECOND\text{-}LCAN = -9$ |
| $r_{b,stom}$ | RES-QLST | $RES\text{-}QLST = r_b$ |
| $r_{b,cut}$ | RES-QLCT | $RES\text{-}QLCT = r_b$ |
| $r_{b,soil}$ | RES-QLSL | $RES\text{-}QLSL = r_b$ |
| $r_{b,lcan}$ | RES-QLLC | $RES\text{-}QLLC = -9$ |
| $r_{dc}$ | RES-CONV | $RES\text{-}CONV = -9$ |


**Table 6.** AQMEII4 dry deposition diagnostic variables for gas phase species corresponding to the resistance framework of Fig. 2b.

**4.2 Bidirectional fluxes of ammonia – a special case**

Some models make use of the concepts of bidirectional fluxes when describing ammonia gas transfer from and to surfaces. In the bidirectional flux paradigm, the difference between the ambient gas concentrations and near-surface (compensation point) concentration is used to determine the direction of the flux: if the ambient air concentration is greater than the compensation point concentration, the flux is downward (i.e. deposition occurs) while in the reverse case the flux is upward (i.e. the emission of ammonia previously stored in the surfaces takes place). The algorithms used in the subset of models employing ammonia bidirectional fluxes were examined, in order to determine common terms that could be used for points of comparison across the algorithms. As an example, we present below (Figure 4 and Table 7) the bidirectional flux model of Nemitz et al. (2001), used within CMAQ to represent bidirectional ammonia gas fluxes. In addition, we also include a comparison of two ammonia bidirectional flux calculations in Appendix C.

The bidirectional flux algorithms were analyzed as a separate case, with the result that a revised and smaller number of variables were reported for the specific case of ammonia bidirectional fluxes than for other gases, focusing on the compensation point concentrations as diagnostics for the cross-comparison of these algorithms. The reported





518 variables in this case are ammonia's aerodynamic resistance, its net surface resistance, and three compensation

519 point concentrations, for stomata, ground and net compensation points, respectively.  These specific parameters for

520 ammonia bidirectional fluxes appear in Table 7, and a detailed comparison of two representative bidirectional

521 ammonia algorithms is presented in Appendix C.

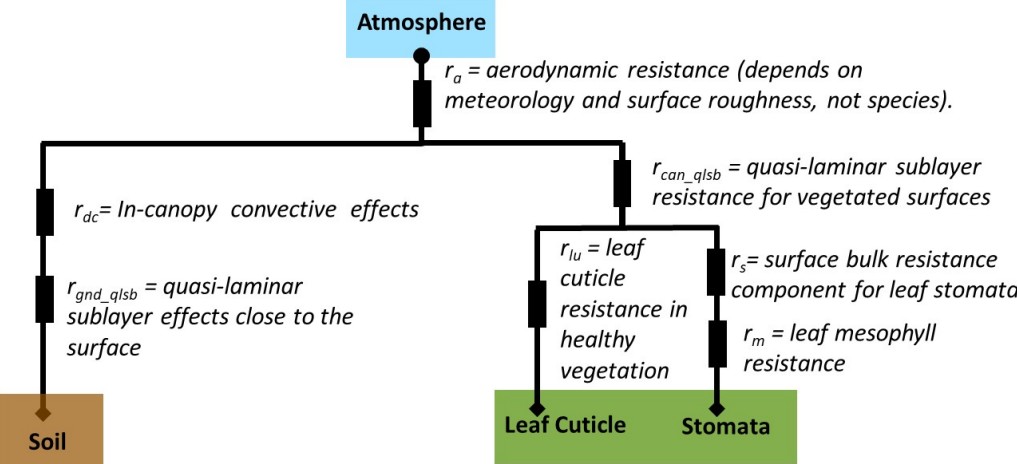

522

**Figure 4.  Nemitz bidirectional flux model for NH₃.**

523

524 In this example, note that the branch containing the $r_{dc}$ term has been designated as the lower canopy pathway, due

525 to the presence of the canopy buoyant convection term $r_{dc}$ (i.e., closest analogy to Wesely's setup is to have the

526 pathway involving deposition to "soil" pathway is designated as a "lower canopy" pathway).

527 Table 7. Variables for bidirectional fluxes of ammonia.

| Name as described here | AQMEII4 Variable Name | Details |
|---|---|---|
| $r_{sum}$ | RES-SUM-NH3 | Net bidirectional flux ammonia resistance |
| $r_a$ | RES-AERO-NH3 | Net Aerodynamic resistance used for ammonia bidirectional fluxes |
| $c_a$ | CONC-NH3-AIR | Air concentration of ammonia used for bidirectional flux calculations |
| $c_c$ | COMP-NH3-NET | Net Ammonia Overall Compensation point concentration |
| $c_g$ | COMP-NH3-GND | Net Ammonia Compensation point concentration with respect to ground |
| $c_s$ | COMP-NH3-STO | Net Ammonia Compensation point concentration with respect to stomata |

528

529

530


**Conclusions**

The fourth phase of the Air Quality Model International Initiative has been introduced. The focus of this phase is on wet and especially dry deposition. The necessity of tackling this subject in a diagnostic way prompted us to divide the initiative in two activities, one dedicated to the evaluation of the process as described by 4-dimensional air quality regional-scale models, the second dealing specifically with evaluating ozone dry deposition calculated by "single-point model" versions of the dry deposition modules used in the regional-scale models with a collection of ozone flux measurements. Here, the organization of Activity 1 has been formally introduced, whereas Activity 2 is presented in a separate companion technical note. In addition to presenting the standard and common input data and the way in which standard output is expected, we also presented the way in which the very diverse representations of dry deposition in participating models have been reduced to a common representation that will facilitate model inter-comparison. The essence of the adopted methodology is the transformation of individual resistances into effective conductances and effective fluxes, which represent the importance of deposition pathways held in common across the models to the total deposition velocity and flux. Resistances held in common across different modelling frameworks were also reported, to allow comparisons at the sub-pathway level, where possible. Thus, regardless of the level of sophistication of the resistance framework, one can meaningfully inter-compare the results produced by different models.

**Data availability.**

No data was generated for this technical note

**Author contributions.**

SG, PM, OEC, and CH led the writing of this technical note. SG, PM, OEC, CH, RB, RB, JB, JD, JF, CDH, IK, DS, and SS conceptualized and implemented the AQMEII4 modeling and analysis framework. JOB, TB, AH, RK, AL, JLPC, JP< YHR, RSJ, MGV, and RW provided documentation of dry deposition schemes used in their models.

**Competing interests.**

The authors declare no conflict of interest

**Acknowledgments**





We gratefully acknowledge the contribution of various groups to the fourth AQMEII activity. The following groups
contributed the data sets used in the grid modeling aspects of this study: U.S. EPA and Environment and Climate
Change Canada (North American emissions processing); TNO (European emissions processing); ECMWF and
Copernicus Atmosphere Monitoring Service (Chemical boundary conditions); ECCAD (archiving and distribution of
the GEIA lightning emissions data based on Price et al. (1997)); Finnish Meteorological Institute (European wildfire
emissions). Ambient North American concentration measurements were extracted from Environment Canada's
National Atmospheric Chemistry Database (NAtChem) PM database and provided by several U.S. and Canadian
agencies (AQS, CAPMoN, CASTNet, IMPROVE, NAPS, SEARCH and CSN networks); North American precipitation-
chemistry measurements were extracted from NAtChem's precipitation-chemistry data base and were provided by
several U.S. and Canadian agencies (CAPMoN, NADP, NBPMN, NSPSN, and REPQ networks); the WMO World Ozone
and Ultraviolet Data Centre (WOUDC) and its data-contributing agencies provided North American and European
ozonesonde profiles; for European air quality data the following data centers were used: EMEP European
Environment Agency/European Topic Center on Air and Climate Change/AirBase provided European air- and
precipitation-chemistry data. Data from meteorological station monitoring networks were obtained from NOAA and
Environment and Climate Change Canada (for the US and Canadian meteorological network data) and the National
Center for Atmospheric Research (NCAR) Data Support Section. The Joint Research Center Ispra/Institute for
Environment and Sustainability provided its ENSEMBLE system for model output harmonization and analyses and
evaluation.
**Disclaimer**
The views expressed in this paper are those of the authors and do not necessarily represent the view or policies of
the U.S. Environmental Protection Agency. This material is based upon work supported, in part, by the National
Center for Atmospheric Research, which is a major facility sponsored by the National Science Foundation under
Cooperative Agreement No. 1852977.

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



## Appendix A: Standard Output Requested From All Participating Models

**Table A1 – AQMEII4 – Meteorology (grid)**

| Variable | Description and Units |
|---|---|
| PRECIP | Sum of all surface precipitation, cm |
| PRESS | Surface pressure, hPa |
| MIXRAT | Water vapour mixing ratio @ 2 m, g kg$^{-1}$ |
| RH | Relative humidity @ 2 m, % |
| TD | Dew point temperature @ 2 m, K |
| TEMP | Air temperature @ 2 m, K |
| WS | Horizontal wind speed @ 10 m, m s$^{-1}$ |
| WD | Horizontal wind direction @ 10 m, deg |
| W | Vertical wind speed @ 10 m, m s$^{-1}$ |
| SWGU | Upward shortwave radiation at the ground, W m$^{-2}$ |
| SWGD | Downward Shortwave Radiation at the ground, W m$^{-2}$ |
| SWTU | Upward shortwave radiation at atmosphere top, W m$^{-2}$ |
| SWTD | Downward shortwave radiation at atmosphere top, W m$^{-2}$ |
| PBL | Planetary boundary layer height, m |
| PAR | Photosynthetically active radiation at the ground, W m$^{-2}$ |
| AOD470 | Aerosol optical depth at 470 nm |
| AOD555 | Aerosol optical depth at 555 nm |
| AOD675 | Aerosol optical depth at 675 nm |
| H2O | Water vapor column, cm3 cm$^{-2}$ |
| USTAR | Friction velocity, m s$^{-1}$ |
| MOL | Monin-Obukhov length, m |
| RHO | Air density of lowest model layer |
| TEMP10 | Air temperature at 10 m, K |
| TSOIL | Uppermost soil layer temperature, K |
| SNOWC | Fractional coverage of snow in grid cell, 0-1 |
| WETCAN | Canopy wetness, 0.0 if dry and 1.0 if wet |
| SOILMOI | Uppermost soil layer moisture, m$^3$ m$^{-3}$ |
| Z0 | Surface roughness length, m |
| ALB | Albedo, fraction |
| Z | Terrain height above sea level, m |
| FWET | Wet surface, unitless fraction |
| LAI-T | Total leaf area index, m$^2$ m$^{-2}$ |




**Table A2. AQMEII4 - Gas and Particle Concentrations and Emissions (grid)**

| Variable | Description and Units |
|---|---|
| SO2 | Concentration of $SO_2$ at ground, $\mu g\ m^{-3}$ |
| NO2 | Concentration of $NO_2$ at ground, $\mu g\ m^{-3}$ |
| NO | Concentration of NO at ground, $\mu g\ m^{-3}$ |
| NOx | Concentration of $NO_x$ at ground, $\mu g\ m^{-3}$ |
| NOy | Concentration of $NO_y$ at ground, $\mu g\ m^{-3}$ |
| HNO3 | Concentration of $HNO_3$ at ground, $\mu g\ m^{-3}$ |
| NH3 | Concentration of $NH_3$ at ground, $\mu g\ m^{-3}$ |
| PAN | Concentration of PAN at ground, $\mu g\ m^{-3}$ |
| HNO4 | Concentration of $HNO_4$ at ground, $\mu g\ m^{-3}$ |
| N2O5 | Concentration of $N_2O_5$ at ground, $\mu g\ m^{-3}$ |
| HONO | Concentration of HONO at ground, $\mu g\ m^{-3}$ |
| ONIT | Concentration of gaseous organic nitrates at ground, $\mu g\ m^{-3}$ |
| O3 | Concentration of $O_3$ at ground, $\mu g\ m^{-3}$ |
| H2O2 | Concentration of $H_2O_2$ at ground, $\mu g\ m^{-3}$ |
| HCHO | Concentration of formaldehyde at ground, $\mu g\ m^{-3}$ |
| CO | Concentration of CO at ground, $\mu g\ m^{-3}$ |
| ETHE | Concentration of ethene at ground, $\mu g\ m^{-3}$ |
| C5H8 | Concentration of isoprene at ground, $\mu g\ m^{-3}$ |
| C10H16 | Concentration of monoterpenes at ground, $\mu g\ m^{-3}$ |
| PM2_5_SU | Concentration of $PM_{2.5}$ Sulphate at ground, $\mu g\ m^{-3}$ |
| PM2_5_AM | Concentration of $PM_{2.5}$ Ammonium at ground, $\mu g\ m^{-3}$ |
| PM2_5_NI | Concentration of $PM_{2.5}$ Nitrate at ground, $\mu g\ m^{-3}$ |
| PM2_5_POA | Concentration of $PM_{2.5}$ Primary Organic Aerosol at ground, $\mu g\ m^{-3}$ |
| PM2_5_SOA | Concentration of $PM_{2.5}$ Secondary Organic Aerosol at ground, $\mu g\ m^{-3}$ |
| PM2_5_OC | Concentration of $PM_{2.5}$ Organic Carbon at ground, $\mu g\ m^{-3}$ |
| PM2_5_EC | Concentration of $PM_{2.5}$ Elemental Carbon (Black Carbon) at ground, $\mu g\ m^{-3}$ |
| PM2_5_SS | Concentration of $PM_{2.5}$ Sea Salt at ground, $\mu g\ m^{-3}$ |
| PM2_5_CA | Concentration of $PM_{2.5}$ Calcium at ground, $\mu g\ m^{-3}$ |
| PM2_5_MG | Concentration of $PM_{2.5}$ Magnesium at ground, $\mu g\ m^{-3}$ |
| PM2_5_NSNA | Concentration of $PM_{2.5}$ Non-Sea-Salt Sodium at ground, $\mu g\ m^{-3}$ |
| PM2_5_PK | Concentration of $PM_{2.5}$ Potassium at ground, $\mu g\ m^{-3}$ |
| PM2_5_FE | Concentration of $PM_{2.5}$ Iron at ground, $\mu g\ m^{-3}$ |





| PM2_5_MN | Concentration of $PM_{2.5}$ Manganese at ground, $\mu g\ m^{-3}$ |
|---|---|
| PM2_5_OTH | Concentration of $PM_{2.5}$ Other (all not speciated) at ground, $\mu g\ m^{-3}$ |
| PM10_SU | Concentration of $PM_{10}$ Sulphate at ground, $\mu g\ m^{-3}$ |
| PM10_AM | Concentration of $PM_{10}$ Ammonium at ground, $\mu g\ m^{-3}$ |
| PM10_NI | Concentration of $PM_{10}$ Nitrate at ground, $\mu g\ m^{-3}$ |
| PM10_POA | Concentration of $PM_{10}$ Primary Organic Aerosol at ground, $\mu g\ m^{-3}$ |
| PM10_SOA | Concentration of $PM_{10}$ Secondary Organic Aerosol at ground, $\mu g\ m^{-3}$ |
| PM10_OC | Concentration of $PM_{10}$ Organic Carbon (at ground, $\mu g\ m^{-3}$ |
| PM10_EC | Concentration of $PM_{10}$ Elemental Carbon (Black Carbon) at ground, $\mu g\ m^{-3}$ |
| PM10_SS | Concentration of $PM_{10}$ Sea Salt at ground, $\mu g\ m^{-3}$ |
| PM10_CA | Concentration of $PM_{10}$ Calcium at ground, $\mu g\ m^{-3}$ |
| PM10_MG | Concentration of $PM_{10}$ Magnesium at ground, $\mu g\ m^{-3}$ |
| PM10_NSNA | Concentration of $PM_{10}$ Non-Sea-Salt Sodium at ground, $\mu g\ m^{-3}$ |
| PM10_PK | Concentration of $PM_{10}$ Potassium at ground, $\mu g\ m^{-3}$ |
| PM10_FE | Concentration of $PM_{10}$ Iron at ground, $\mu g\ m^{-3}$ |
| PM10_MN | Concentration of $PM_{10}$ Manganese at ground, $\mu g\ m^{-3}$ |
| PM10_OTH | Concentration of $PM_{10}$ Other (all not speciated) at ground, $\mu g\ m^{-3}$ |
| PMTOT_SU | Concentration of PMTOT Sulphate at ground, $\mu g\ m^{-3}$ |
| PMTOT_AM | Concentration of PMTOT Ammonium at ground, $\mu g\ m^{-3}$ |
| PMTOT_NI | Concentration of PMTOT Nitrate at ground, $\mu g\ m^{-3}$ |
| PMTOT_POA | Concentration of PMTOT Primary Organic Aerosol at ground, $\mu g\ m^{-3}$ |
| PMTOT_SOA | Concentration of PMTOT Secondary Organic Aerosol at ground, $\mu g\ m^{-3}$ |
| PMTOT_OC | Concentration of PMTOT Organic Carbon at ground, $\mu g\ m^{-3}$ |
| PMTOT_EC | Concentration of PMTOT Elemental Carbon (Black Carbon) at ground, $\mu g\ m^{-3}$ |
| PMTOT_SS | Concentration of PMTOT Sea Salt at ground, $\mu g\ m^{-3}$ |
| PMTOT_CA | Concentration of PMTOT Calcium at ground, $\mu g\ m^{-3}$ |
| PMTOT_MG | Concentration of PMTOT Magnesium at ground, $\mu g\ m^{-3}$ |
| PMTOT_NSNA | Concentration of PMTOT Non-Sea-Salt Sodium at ground, $\mu g\ m^{-3}$ |
| PMTOT_PK | Concentration of PMTOT Potassium at ground, $\mu g\ m^{-3}$ |
| PMTOT_FE | Concentration of PMTOT Iron at ground, $\mu g\ m^{-3}$ |
| PMTOT_MN | Concentration of PMTOT Manganese at ground, $\mu g\ m^{-3}$ |
| PMTOT_OTH | Concentration of PMTOT Other (all not speciated) at ground, $\mu g\ m^{-3}$ |
| PM2_5 | Concentration of $PM_{2.5}$ at ground, $\mu g\ m^{-3}$ |
| PM2_5N | Number concentration of $PM_{2.5}$ at ground, $cm^{-3}$ |
| PM10 | Concentration of $PM_{10}$ at ground, $\mu g\ m^{-3}$ |
| PM10N | Number concentration of $PM_{10}$ at ground, $cm^{-3}$ |



| | |
|---|---|
| PMTOT | Concentration of total PM at ground, $\mu g\ m^{-3}$ |
| PMTOTN | Number concentration of total PM at ground, $cm^{-3}$ |
| JNO2 | Photolysis rate of $NO_2$ at ground, $1E\text{-}3\ s^{-1}$ |
| E_SO2 | Accumulated emission of $SO_2$, $kg\ km^{-2}$ |
| E_ANOX | Accumulated emission of anthropogenic $NO+NO_2$ as $NO_2$, $kg\ km^{-2}$ |
| E_NH3 | Accumulated emission of $NH_3$, $kg\ km^{-2}$ |
| E_CO | Accumulated emission of CO, $kg\ km^{-2}$ |
| E_PM2_5 | Accumulated emission of primary $PM_{2.5}$, $kg\ km^{-2}$ |
| E_PM10 | Accumulated emission of primary $PM_{10}$, $kg\ km^{-2}$ |
| E_ETHE | Accumulated emission of ethene, $kg\text{-}C\ km^{-2}$ |
| E_TOLU | Accumulated emission of toluene, $kg\text{-}C\ km^{-2}$ |
| E_HCHO | Accumulated emission of formaldehyde, $kg\text{-}C\ km^{-2}$ |
| E_C5H8 | Accumulated emission of isoprene, $kg\text{-}C\ km^{-2}$ |
| E_MNTP | Accumulated emission of monoterpenes, $kg\text{-}C\ km^{-2}$ |
| E_SQTP | Accumulated emission of sesquiterpenes, $kg\text{-}C\ km^{-2}$ |
| E_OVOC | Accumulated emission other VOCs not in above groups, $kg\text{-}C\ km^{-2}$ |
| E_SNOX | Accumulated emission of soil $NO+NO_2$ as $NO_2$, $kg\ km^{-2}$ |
| E_SS | Accumulated emission of sea salt (all particle sizes), $kg\ km^{-2}$ |
| E_WBDUST | Accumulated emission of wind blown dust (all particle sizes), $kg\ km^{-2}$ |
| PM2_5_WAT | Concentration of $PM_{2.5}$ water at ground (if calculated), $\mu g\ m^{-3}$ |
| PM10_WAT | Concentration of $PM_{10}$ water at ground (if calculated), $\mu g\ m^{-3}$ |
| PMTOT_WAT | Concentration of PMTOT water at ground (if calculated), $\mu g\ m^{-3}$ |







**Table A3. AQMEII4 – Deposition Fluxes (grid)**

| | |
|---|---|
| WFLUX-HSO3- | Wet deposition flux of $HSO_3^-$ ion, eq ha$^{-1}$ |
| WFLUX-SO4= | Wet deposition flux of $SO_4^=$ ion, eq ha$^{-1}$ |
| WFLUX-NO3- | Wet deposition flux of $NO_3^-$ ion, eq ha$^{-1}$ |
| WFLUX-NH4+ | Wet deposition flux of $NH_4^+$ ion, eq ha$^{-1}$ |
| WFLUX-BCT1 | Wet deposition flux of base cations, eq ha$^{-1}$ |
| WFLUX-TOC | Wet deposition flux of total organic carbon, g ha$^{-1}$ |
| PRECIP | Surface precipitation, cm |
| DFLUX-SO2 | Dry deposition flux of sulphur dioxide gas, eq ha$^{-1}$ |
| DFLUX-NO2 | Dry deposition flux of nitrogen dioxide gas, eq ha$^{-1}$ |
| DFLUX-NO | Dry deposition flux of nitrogen monoxide gas, eq ha$^{-1}$ |
| DFLUX-HNO3 | Dry deposition flux of nitric acid gas, eq ha$^{-1}$ |
| DFLUX-NH3 | Net flux of ammonia gas (negative if upwards), eq ha$^{-1}$ |
| DFLUX-PAN | Dry deposition flux of peroxyacetylnitrate gas, eq ha$^{-1}$ |
| DFLUX-HNO4 | Dry deposition flux of peroxynitric acid gas, eq ha$^{-1}$ |
| DFLUX-N2O5 | Dry deposition flux of dinitrogen pentoxide gas, eq ha$^{-1}$ |
| DFLUX-ONIT | Dry deposition flux of gaseous organic nitrate, eq ha$^{-1}$ |
| DFLUX-O3 | Dry deposition flux of ozone gas, g ha$^{-1}$ |
| DFLUX-H2O2 | Dry deposition flux of hydrogen peroxide gas, g ha$^{-1}$ |
| DFLUX-HCHO | Dry deposition flux of formaldehyde gas, g ha$^{-1}$ |
| DFLUX-P-SO4 | Dry deposition flux of total particle sulphate, eq ha$^{-1}$ |
| DFLUX-P-NO3 | Dry deposition flux of total particle nitrate, eq ha$^{-1}$ |
| DFLUX-P-NH4 | Dry deposition flux of total particle ammonium, eq ha$^{-1}$ |
| DFLUX-P-TC | Dry deposition flux of total particle organic carbon, g ha$^{-1}$ |
| DFLUX-P-EC | Dry deposition flux of total black carbon, g ha$^{-1}$ |
| DFLUX-P-BCT1 | Dry deposition flux of total particulate base cations, eq ha$^{-1}$ |
| DFLUX-P-BCT2 | Flux of base cat. removed as non-transportable fraction during emissions processing (if available), eq ha$^{-1}$ |



| DFLUX-P-SS | Dry deposition flux of total sea salt aerosol, moles ha$^{-1}$ |
| --- | --- |
| DFLUX-P-CM | Dry deposition flux of total crustal material (all particulate components not speciated above), g ha$^{-1}$ |
| DFLUX-PM2_5 | Dry deposition flux of PM$_{2.5}$, g ha$^{-1}$ |
| DFLUX-HONO | Dry deposition flux of HONO, eq ha$^{-1}$ |
| RES-AERO | Aerodynamic resistance, s cm$^{-1}$ |




**Appendix B: Resistance Diagrams and Calculation of AQMEII4 Reported Dry Deposition Diagnostic Variables for Dry Deposition Schemes Implemented in Participating Models**

***Example 1: GEM-MACH model, default Robichaud scheme.***

These are the calculations for the Environment and Climate Change Canada model GEM-MACH (Global Environmental Multiscale- Modelling Air-quality and CHemistry). The resistance diagram for this model is shown in Figure B1. The deposition algorithm closely follows Wesely's original hence the similarities to Figure 1. The scheme includes further modifications incorporating parameterizations from Jarvis (1976), Val Martin et al. (2014) and other authors; details and references for this scheme may be found in Makar et al (2018) , Supplemental Information). In GEM-MACH, snow, when present, is treated as a separate land use type.

Figure B1. Resistance diagram for the ECCC GEM-MACH model (default Robichaud scheme).

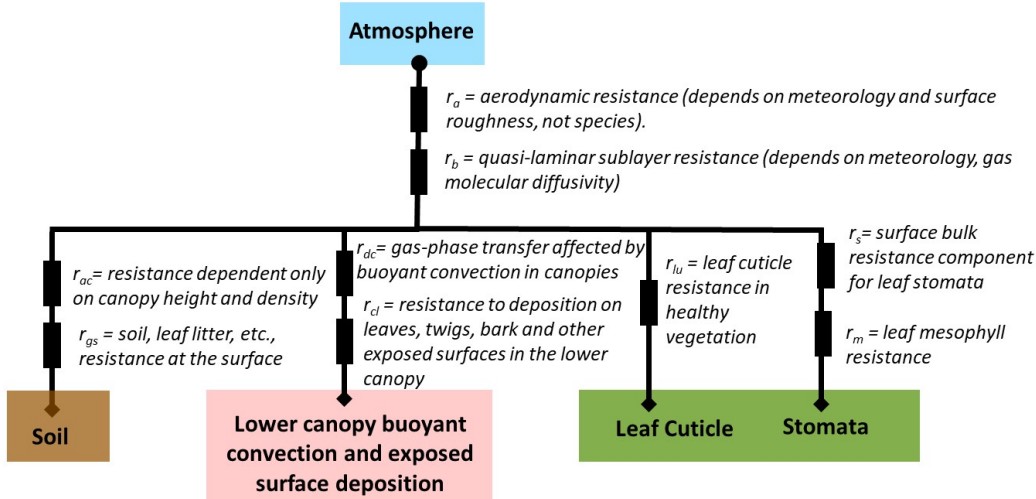

The main difference between the resistances in Wesely (1989) and the GEM-MACH resistances (aside from formulation details) is the addition of a surface wetness term, (1-Wst), intended to account for the influence of wet surfaces on dry deposition.

Table B1. Example 1: AQMEII4 reported gaseous deposition variables corresponding to the GEM-MACH/Robichaud resistance model of Figure B1.



| Name as described here | AQMEII4 Variable Name | Formulae |
|---|---|---|
| $r_a$ | RES-AERO | $RES\text{-}AERO = r_a$ |
| $r_c$ | RES-SURF | $RES\text{-}SURF = \left((1-W_{st})(r_s+r_m)^{-1} + (r_{lu})^{-1} + (r_{dc}+r_{cl})^{-1} + (r_{ac}+r_{gs})^{-1}\right)^{-1}$ |
| $r_s$ | RES-STOM | $RES\text{-}STOM = r_s$ |
| $r_m$ | RES-MESO | $RES\text{-}MESO = r_m$ |
| $r_c$ | RES-CUT | $RES\text{-}CUT = r_{lu}$ |
| $E_{STOM}$ | ECOND-ST | $ECOND\text{-}ST = \left(\dfrac{(1-W_{st})(r_s+r_m)^{-1}}{(1-W_{st})(r_s+r_m)^{-1}+(r_{lu})^{-1}+(r_{dc}+r_{cl})^{-1}+(r_{ac}+r_{gs})^{-1}}\right)V_d$ |
| $E_{CUT}$ | ECOND-CUT | $ECOND\text{-}CUT = \left(\dfrac{(r_{lu})^{-1}}{(1-W_{st})(r_s+r_m)^{-1}+(r_{lu})^{-1}+(r_{dc}+r_{cl})^{-1}+(r_{ac}+r_{gs})^{-1}}\right)V_d$ |
| $E_{SOIL}$ | ECOND-SOIL | $ECOND\text{-}SOIL = \left(\dfrac{(r_{dc}+r_{cl})^{-1}}{(1-W_{st})(r_s+r_m)^{-1}+(r_{lu})^{-1}+(r_{dc}+r_{cl})^{-1}+(r_{ac}+r_{gs})^{-1}}\right)V_d$ |
| $E_{LCAN}$ | ECOND-LCAN | $ECOND\text{-}LCAN = \left(\dfrac{(r_{dc}+r_{cl})^{-1}}{(1-W_{st})(r_s+r_m)^{-1}+(r_{lu})^{-1}+(r_{dc}+r_{cl})^{-1}+(r_{ac}+r_{gs})^{-1}}\right)V_d$ |
| $r_{b,\,stom}$ | RES-QLST | $RES\text{-}QLST = r_b$ |
| $r_{b,cut}$ | RES-QLCT | $RES\text{-}QLCT = r_b$ |
| $r_{b,soil}$ | RES-QLSL | $RES\text{-}QLSL = r_b$ |
| $r_{b,lcan}$ | RES-QLLC | $RES\text{-}QLLC = r_b$ |
| $r_{dc}$ | RES-CONV | $RES\text{-}CONV = r_{dc}$ |







*Example 2: CMAQ M3DRY.*
The second specific air-quality model example is the M3DRY algorithm implemented in the US EPA's
Community Multiscale Air Quality (CMAQ) model, one of two available dry deposition options in that
model.   In this particular case, separate branches occur for the vegetated versus non-vegetated fraction
within each model grid cell, and further branching resistance pathways take into account the fraction of
the grid cell which is wet versus dry, and snow-covered versus non-snow covered.  In-canopy convective
effects are only calculated for the vegetated fraction.
Figure B2.  Resistance diagram for the US EPA CMAQ model with the M3DRY deposition option.

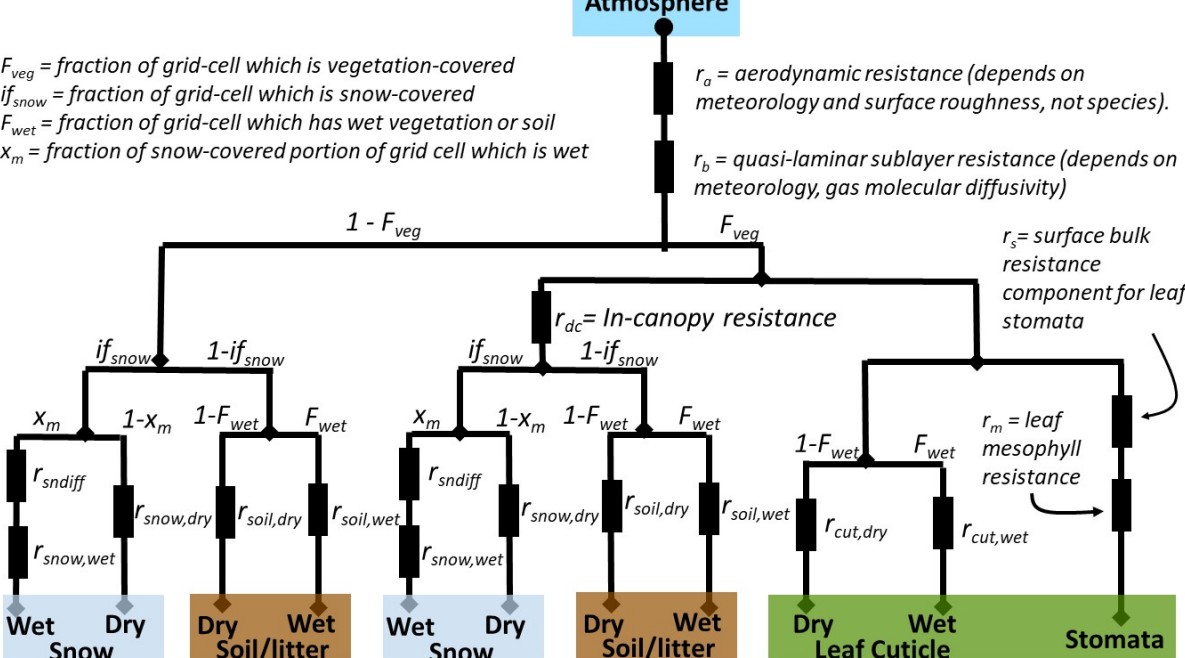














Table B2. AQMEII4 reported gaseous deposition variables corresponding to the CMAQ M3Dry resistance
model of Figure B2.

| Name as described here | AQMEII4 Variable Name | Formulae |
|---|---|---|
| $r_a$ | RES-AERO | $RES\text{-}AERO = r_a$ |
| $r_c$ | RES-SURF | $RES\text{-}SURF =$ $\left\{ F_{veg}\left( \frac{1}{r_s+r_m} + \frac{(1-F_{wet})LAI}{r_{cut,dry}} + \frac{F_{wet}*LAI}{r_{cut,wet}} + \frac{1}{r_{dc}+\frac{1}{(1-ifsnow)\left(\frac{(1-F_{wet})}{r_{soil,dry}}+\frac{F_{wet}}{r_{soil,wet}}\right)+(ifsnow)\left(\frac{(1-x_m)}{r_{snow,dry}}+\frac{1}{r_{sndiff}}\right)}} \right) \right.$ $+ (1-F_{veg})\left( (1-ifsnow)\left(\frac{(1-F_{wet})}{r_{soil,dry}}+\frac{F_{wet}}{r_{soil,wet}}\right) + (ifsnow)\left(\frac{(1-x_m)}{r_{snow,dry}}+\frac{x_m}{r_{sndiff}+r_{sno}}\right) \right.$ |
| $r_s$ | RES-STOM | $RES\text{-}STOM = r_s$ |
| $r_m$ | RES-MESO | $RES\text{-}MESO = r_m$ |
| $r_c$ | RES-CUT | $RES\text{-}CUT = \left[ \left( \frac{(1-F_{wet})LAI}{r_{cut,dry}} + \frac{F_{wet}*LAI}{r_{cut,wet}} \right) \right]^{-1}$ |
| $E_{STOM}$ | ECOND-ST | $ECOND\text{-}ST = \left[ \frac{(F_{veg})}{(r_s+r_m)} \right] (RES-SURF)\, V_d$ |
| $E_{CUT}$ | ECOND-CUT | $ECOND\text{-}CUT = (RES-CUT)^{-1}(RES-SURF)V_d$ |
| $E_{SOIL}$ | ECOND-SOIL | $ECOND\text{-}SOIL = \left[ (1-F_{veg})\left( (1-ifsnow)\left(\frac{(1-F_{wet})}{r_{soil,dry}}+\frac{F_{wet}}{r_{soil,wet}}\right) + (ifsnow)\left(\frac{(1-x_m)}{r_{snow,dry}}+\frac{x_m}{r_{sndiff}+r_{snow,wet}}\right) \right) \right] (RES-SURF)\, V_d$ |
| $E_{LCAN}$ | ECOND-LCAN | $ECOND\text{-}LCAN = \left[ \frac{F_{veg}}{r_{dc}+\frac{1}{(1-ifsnow)\left(\frac{(1-F_{wet})}{r_{soil,dry}}+\frac{F_{wet}}{r_{soil,wet}}\right)+(ifsnow)\left(\frac{(1-x_m)}{r_{snow,dry}}+\frac{x_m}{r_{sndiff}+r_{snow,wet}}\right)}} \right] (RES-SURF)\, V_d$ |
| $r_{b,stom}$ | RES-QLST | $RES\text{-}QLST = r_b$ |
| $r_{b,cut}$ | RES-QLCT | $RES\text{-}QLCT = r_b$ |
| $r_{b,soil}$ | RES-QLSL | $RES\text{-}QLSL = r_b$ |
| $r_{b,lcan}$ | RES-QLLC | $RES\text{-}QLLC = r_b$ |
| $r_{dc}$ | RES-CONV | $RES\text{-}CONV = r_{dc}$ |

Note that the vegetated fraction and leaf area index used in the above equations for CMAQ with the M3DRY
deposition option is for specific LULC types: the quantities in Table B2 will be reported for each of the 16 generic
LULC categories for AQMEII4. Note that the lower canopy pathway has been identified as such due to the presence
of the $r_{dc}$ term; i.e. this points to its similarity with Wesely's original lower canopy pathway.


***Example 3: CMAQ STAGE.***
The third specific air-quality model example is the algorithm used by the US EPA's Community Multiscale
Air Quality (CMAQ) model with the Surface Tiled Aerosol and Gaseous Exchange (STAGE) deposition
option. In this particular case, separate branches occur for the vegetated versus non-vegetated fraction
for each LULC type within each model grid cell, and further branching resistance pathways take into
account the fraction of the grid cell which is wet versus dry, and snow-covered versus non-snow covered.
In-canopy convective effects are only calculated for in the vegetated fraction.
Figure B3.  Resistance diagram for the US EPA CMAQ model with the STAGE deposition option. Note, that
this is an extension of the Massad et al. 2010 and Nemitz et al. 2001 resistance model in the CMAQ
modeling framework.

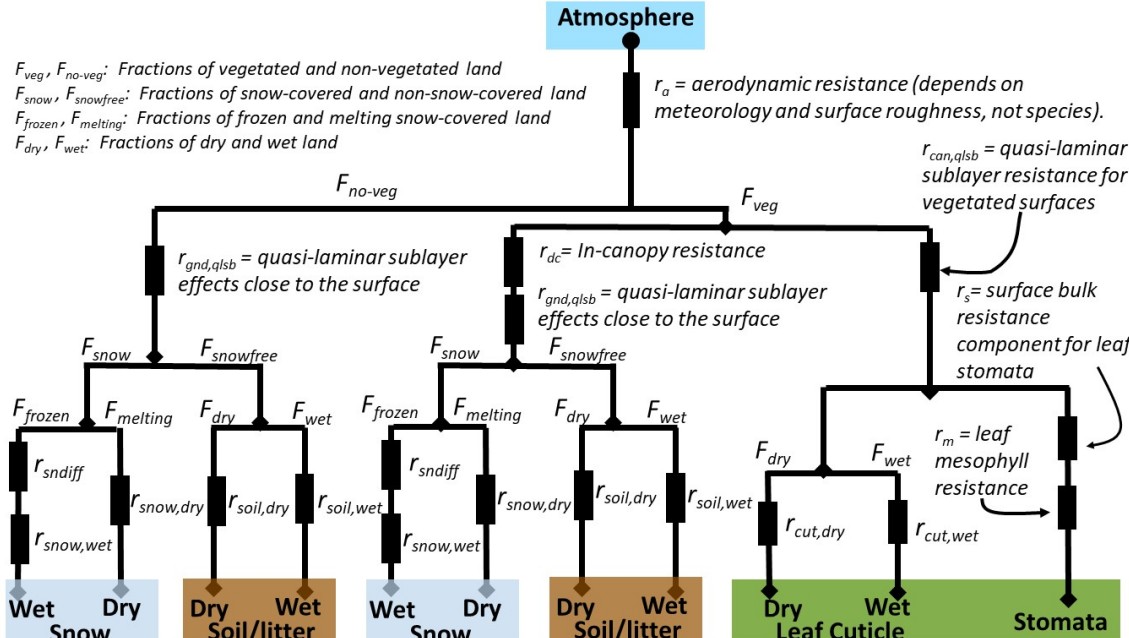















Table B3. AQMEII4 reported gaseous deposition variables corresponding to the CMAQ STAGE resistance
model of Figure B3.

| Name as described here | AQMEII4 Variable Name | Formulae |
|---|---|---|
| $r_a$ | RES-AERO | $RES\text{-}AERO = r_a$ |
| $r_c$ | RES-SURF | $RES\text{-}SURF = \left( \left( r_{can,qlsb} + ((r_s+r_m)^{-1} + (r_{cut})^{-1})^{-1} \right)^{-1} + \left( r_{dc} + r_{gnd,qlsb} + r_{soil} \right)^{-1} \right)^{-1}$ |
| $r_s$ | RES-STOM | $RES\text{-}STOM = r_s$ |
| $r_m$ | RES-MESO | $RES\text{-}MESO = r_m$ |
| $r_c$ | RES-CUT | $RES\text{-}CUT = r_{cut}$ |
| $E_{STOM}$ | ECOND-ST | $ECOND\text{-}ST = \left[ \frac{(F_{veg})}{(r_s+r_m)} \right] (RES-SURF)\, V_d$ |
| $E_{CUT}$ | ECOND-CUT | $ECOND\text{-}CUT = \left[ \frac{F_{veg}}{r_{cut}} \right] (RES-SURF) V_d$ |
| $E_{SOIL}$ | ECOND-SOIL | $ECOND\text{-}SOIL = \left[ \frac{F_{no\,veg}}{r_{gnd,qlsb}+r_{soil}} \right] (RES-SURF)\, V_d$ |
| $E_{LCAN}$ | ECOND-LCAN | $ECOND\text{-}LCAN = \left[ \frac{F_{veg}}{r_{dc}+r_{gnd,qlsb}+r_{soil}r_{dc}+\frac{1}{r_{soil}}} \right] (RES-SURF)\, V_d$ |
| $r_{b,\,stom}$ | RES-QLST | $RES\text{-}QLST = r_{can,qlsb}$ |
| $r_{b,cut}$ | RES-QLCT | $RES\text{-}QLCT = r_{can,qlsb}$ |
| $r_{b,soil}$ | RES-QLSL | $RES\text{-}QLSL = r_{gnd,qlsb}$ |
| $r_{b,lcan}$ | RES-QLLC | $RES\text{-}QLLC = r_{gnd,qlsb}$ |
| $r_{dc}$ | RES-CONV | $RES\text{-}CONV = r_{dc}$ |

Where
$F_{veg} + F_{no\,veg} = 1$      Vegetation coverage fractions
$F_{snow} + F_{snowfree} = 1$      Snow coverage fraction
$F_{wet} + F_{dry} = 1$      Surface wetness fractions
$F_{frozen} + F_{melting} = 1$      Snow melt fractions
$r_{cut=} \left( LAI \left( \frac{F_{dry}}{r_{cut,dry}} + \frac{F_{wet}}{r_{cut,wet}} \right) \right)^{-1}$
$r_{soil} = \left( F_{no\,snow} \left( \frac{F_{dry}}{r_{soil,dry}} + \frac{F_{wet}}{r_{soil,wet}} \right) + F_{snow} \left( \frac{F_{frozen}}{r_{snow,dry}} + \frac{F_{melting}}{r_{sndiff} + r_{snow,wet}} \right) \right)^{-1}$
Note that the vegetated fraction and leaf area index used in the above equations for CMAQ with the STAGE
deposition option is for specific LULC types: the quantities in Table B3 will be reported for each of the 16 generic
LULC categories for AQMEII4. Note that the lower canopy pathway has been identified as such due to the presence
of the $r_{dc}$ term; i.e. this points to its similarity with Wesely's original lower canopy pathway.






**Example 4. LOTOS EUROS**
Figure B4. Resistance diagram for the dry deposition scheme implemented in LOTOS EUROS

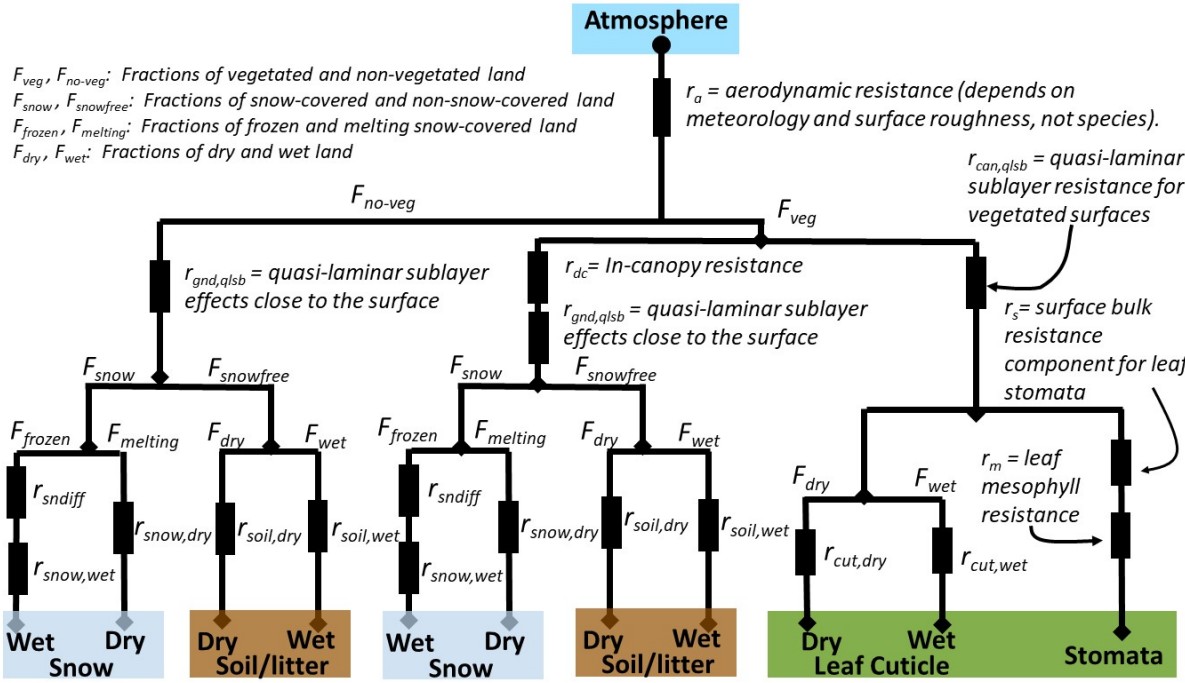













Table B4. AQMEII4 reported gaseous deposition variables corresponding to the LOTOS-EUROS resistance
model of Figure B4



| Name as described here | AQMEII4 Variable Name | Description | Formulae |
|---|---|---|---|
| $R_a$ | RES_AERO | Aerodynamic resistance | $RES\_AERO = \dfrac{\ln\left(\frac{z_r}{z_0}\right) + 4.7\left(\frac{z_r - z_0}{L}\right)}{\kappa \cdot u^*}$ for stable conditions, $\kappa$: von Karman constant (here 0.35), $L$: Monin-Obukhov length, $z_r$: reference height, $z_0$: height of surface roughness |
| $R_b$ | RES_QLSB | Quasi-laminar sublayer resistance | $RES\_QLSB = 1.3 \cdot 150 \cdot \sqrt{\dfrac{L_d}{V(h)}}$, $L_d$: cross-wind lead dimension, $V(h)$: wind speed at canopy top $h$, factor 1.3 accounts for differences in diffusivity between heat and ozone |
| $R_c$ | RES_SURF | Net canopy resistance | $RES_{SURF} = \left(\dfrac{1}{R_w} + \dfrac{1}{R_{inc} + R_{soil}} + \dfrac{1}{R_s}\right)^{-1}$ for $NO_2$, $NH_3$, $SO_2$, $O_3$ <br> $RES_{SURF} = 10$; $RES_{SURF} = 50$ (wet conditions) for HNO3, N2O5, NO3, H2O2 <br> $RES_{SURF} = 2000$ (wet condition); $RES_{SURF} = 500 - 70$ (snow condition); $RES_{surf} = 9999$ (other conditions) for NO, CO |
| $R_w$ | RES_CUT | Net cuticle resistance | $RES_{CUT} = 2000$ for NO2 <br> $RES_{CUT} = 2500$ for O3 <br> $RES_{CUT} = 25000 * e^{(-0.0693 * rh)}$ for SO2 if rh $< 81.3$ <br> $RES_{CUT} = 5.8 * 10^{11} * e^{(-0.278 * rh)}$ for SO2 if rh $> 81.3$ <br> $RES_{CUT} = SAI \cdot a \cdot e^{(100 - RH)/\beta}$ for NH3 <br> $SAI$: surface area index, $a$=2 s/m, $\beta$=12, $RH$: relative humidity (%) |
| $R_{inc}$ | | In canopy resistance | $RES\_LCAN = \dfrac{b \cdot h \cdot SAI}{u^*}$, $b$: empirical constant (14 m$^{-1}$), $h$: height of vegetation (m), $SAI$: surface area index, $u^*$: friction velocity (m s$^{-1}$) |
| $R_{soil}$ | RES_SOIL | Soil resistance | Parametrized, frozen soil, wet soil, dry soil <br> RES_SOIL_FROZEN=1000 s m$^{-1}$ for NH3; 2000 s m$^{-1}$ for O3, NO2; 500 s m$^{-1}$ for SO2 <br> RES_SOIL_WET = 10 s m$^{-1}$ for NH3, SO2; 2000 s m$^{-1}$ for O3, NO2 <br> RES_SOIL_DRY (landuse dependent) 200-2000 s m$^{-1}$ for O3; 10-100 s m$^{-1}$ for NH3; 10-1000 s m$^{-1}$ for SO2; 1000-2000 s m$^{-1}$ for NO2 |
| $R_s$ | RES_STOM | Net stomatal resistance | $RES\_STOM = \dfrac{1}{E_{stom}}$ |
| $E_{STOM}$ | ECOND_ST | Effective conductance associated with deposition to plant stomata | $ECOND_{ST} = EMax_{stom} * F_{light} * F_{phen} * F_{temp} * F_{vpd} * F_{swp} * C_{diff}$ <br> EMax: Maximum stomatal conductance (derived for ozone, landuse dependent) <br> F_light, F_phen, F_temp, F_vpd, F_swp: Factors [0-1] for conductance dependency of light, phenology, temperature, vapour pressure and soil-water <br> C_diff: Diffusion coefficient for species with respect to ozone <br> Mesophyll conductance part incorporated in Stomatal conductance |





| C_comp | | Bidirectional fluxes of NH$_3$ | Use of compensation point to derive bi-directional flux for NH3 following:<br><br>Wichink Kruit et al, Modeling the distribution of ammonia across Europe including bi-directional surface–atmosphere exchange. https://doi.org/10.5194/bg-9-5261-2012 |
|---|---|---|---|








*Example 5:  GEM-MACH model, Zhang scheme.*
These are the calculations for the Environment and Climate Change Canada model GEM-MACH (Global
Environmental Multiscale- Modelling Air-quality and CHemistry), using the scheme of Zhang et al (2003,
2010). The resistance diagram for this model is shown in Figure B5.
Figure B5.  Resistance diagram for the ECCC GEM-MACH model (Zhang scheme).

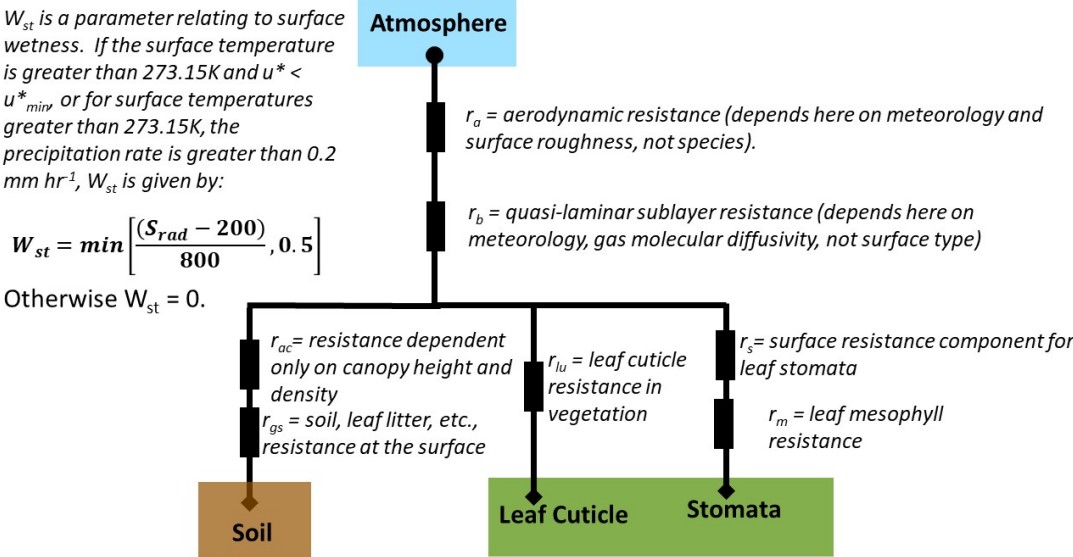


The main difference in the overall construction of the deposition scheme relative to the default Robichaud
scheme (aside from the details of how the different terms are calculated) is in the absence of the lower
canopy buoyant convection and exposed surface deposition branch of Wesely's original model.  The
details of the parameterizations for the terms in the equations also differ from the Robichaud scheme.


Table B5.  AQMEII4 reported gaseous deposition variables corresponding to the GEM-MACH/Zhang
resistance model of Figure B5.

| *Name as described here* | AQMEII4 Variable Name | Formulae |
|---|---|---|
| $r_a$ | RES-AERO | $RES\text{-}AERO = r_a$ |
| $r_c$ | RES-SURF | $RES\text{-}SURF = \left((1 - W_{st})(r_s + r_m)^{-1} + (r_{lu})^{-1} + \left(r_{ac} + r_{gs}\right)^{-1}\right)^{-1}$ |
| $r_s$ | RES-STOM | $RES\text{-}STOM = r_s$ |
| $r_m$ | RES-MESO | $RES\text{-}MESO = r_m$ |
| $r_c$ | RES-CUT | $RES\text{-}CUT = r_{lu}$ |




| $E_{STOM}$ | ECOND-ST | $ECOND\text{-}ST = \left( \dfrac{(1-W_{st})(r_s+r_m)^{-1}}{(1-W_{st})(r_s+r_m)^{-1}+(r_{lu})^{-1}+(r_{ac}+r_{gs})^{-1}} \right) V_d$ |
|---|---|---|
| $E_{CUT}$ | ECOND-CUT | $ECOND\text{-}CUT = \left( \dfrac{(r_{lu})^{-1}}{(1-W_{st})(r_s+r_m)^{-1}+(r_{lu})^{-1}+(r_{ac}+r_{gs})^{-1}} \right) V_d$ |
| $E_{SOIL}$ | ECOND-SOIL | $ECOND\text{-}SOIL = \left( \dfrac{(r_{dc}+r_{cl})^{-1}}{(1-W_{st})(r_s+r_m)^{-1}+(r_{lu})^{-1}+(r_{ac}+r_{gs})^{-1}} \right) V_d$ |
| $E_{LCAN}$ | ECOND-LCAN | $ECOND\text{-}LCAN = -9$ |
| $r_{b,stom}$ | RES-QLST | $RES\text{-}QLST = r_b$ |
| $r_{b,cut}$ | RES-QLCT | $RES\text{-}QLCT = r_b$ |
| $r_{b,soil}$ | RES-QLSL | $RES\text{-}QLSL = r_b$ |
| $r_{b,lcan}$ | RES-QLLC | $RES\text{-}QLLC = r_b$ |
| $r_{dc}$ | RES-CONV | $RES\text{-}CONV = -9$ |





**Example 6. WRF-Chem**
Figure B6. Resistance diagram for the gaseous dry deposition scheme implemented in WRF-Chem

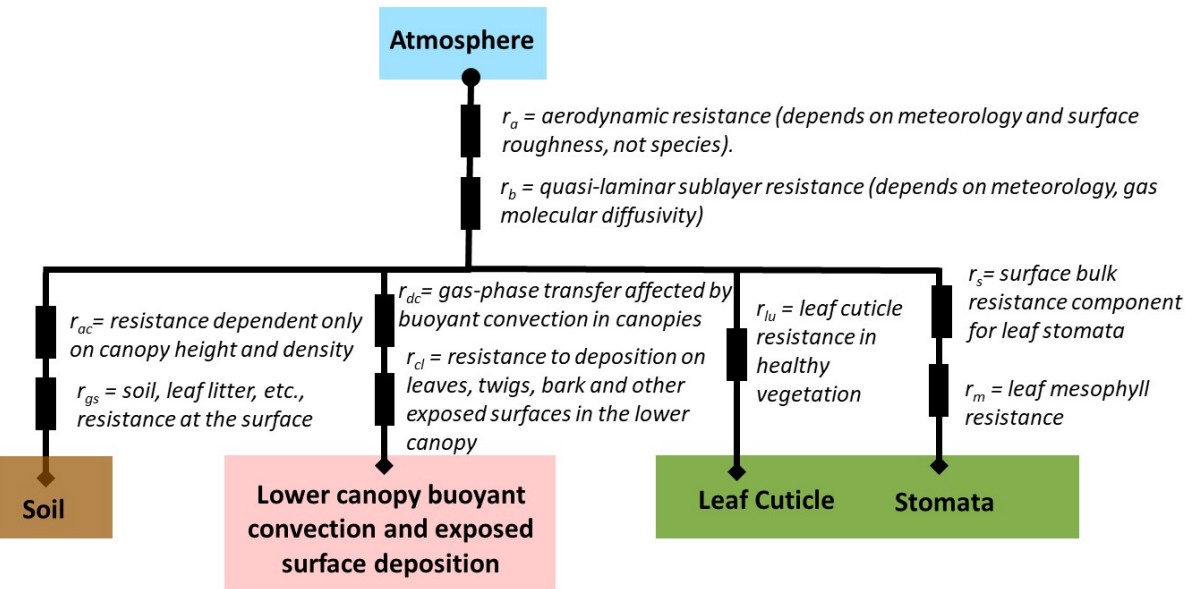


Table B6. AQMEII4 reported gaseous deposition variables corresponding to the WRF-Chem resistance model of
Figure B6.






| Name | AQMEII4 Name | Description | Formula |
|---|---|---|---|
| $V_d$ | VD | Deposition velocity | $$Vd = \frac{1}{r_a + r_b + r_c}$$ |
| $r_a$ | RES-AERO | Aerodynamic resistance | Stable: $\quad r_a = \dfrac{0.74\,ln(\frac{z}{z_0}) + 4.7\frac{z - z_0}{L}}{ku^*} \qquad z = 2m.$<br><br>Neutral: $\quad r_a = \dfrac{0.74\,ln(\frac{z}{z_0})}{ku^*} \qquad z = 2m.$<br><br>Unstable: $\quad r_a = \dfrac{0.74}{ku^*}\left\{ln\left[\dfrac{\sqrt{1 - 9\frac{z}{L}} - 1}{\sqrt{1 - 9\frac{z}{L}} + 1}\right] - ln\left[\dfrac{\sqrt{1 - 9\frac{z_0}{L}} - 1}{\sqrt{1 - 9\frac{z_0}{L}} + 1}\right]\right\}$ |
| $r_c$ | RES-SURF | Bulk surface resistance | $$r_c = \frac{1}{\frac{1}{r_m + r_s} + \frac{1}{r_{cut}} + \frac{1}{r_{dc} + r_{cl}} + \frac{1}{r_{ac} + r_{gs}}}$$ |
| $r_s$ | RES-STOM | Net stomatal resistance | $$r_s = ri\left\{1 + \left(\frac{200}{Rad + 0.1}\right)^2\right\}\frac{400}{T(40 - T)}$$ |
| $r_m$ | RES-MESO | Net mesophyll resistance | $$r_m = \frac{1}{\frac{H}{3000} + 100 f_i}$$ |
| $r_{cut}$ | RES-CUT | Net cuticle resistance | $$r_{cut} = r_{lu}$$ |
| $E_{STOM}$ | ECOND-ST | Effective conductance associated with deposition to plant stomata | $$E_{STOM} = \frac{1}{r_m + r_s} r_c V_d$$ |
| $E_{CUT}$ | ECOND-CUT | Effective conductance associated with deposition to plant cuticles | $$E_{CUT} = \frac{1}{r_{cut}} r_c V_d$$ |
| $E_{SOIL}$ | ECOND-SOIL | Effective conductance associated with deposition to soil and un-vegetated surfaces | $$E_{SOIL} = \frac{1}{r_{ac} + r_{gs}} r_c V_d$$ |





| $E_{LCAN}$ | ECOND-LCAN | Effective conductance associated with deposition to the lower canopy. | $E_{LCAN} = \dfrac{1}{r_{dc} + r_{cl}} r_c V_d$ |
|---|---|---|---|
| $r_{b, stom}$ | RES-QLST | RES_QLST= $r_b$ Quasi-laminar sub-layer resistance | $r_b = 2(ku^*)^{-1}(S_c/P_r)^{2/3}$ |
| $r_{b,cut}$ | RES-QLCT | RES_QLCT= $r_b$ Quasi-laminar sub-layer resistance | $r_b = 2(ku^*)^{-1}(S_c/P_r)^{2/3}$ |
| $r_{b,soil}$ | RES-QLSL | RES_QLSL= $r_b$ Quasi-laminar sub-layer resistance | $r_b = 2(ku^*)^{-1}(S_c/P_r)^{2/3}$ |
| $r_{b,lcan}$ | RES-QLLC | RES_QLLC= $r_b$ Quasi-laminar sub-layer resistance | $r_b = 2(ku^*)^{-1}(S_c/P_r)^{2/3}$ |
| $r_{dc}$ | RES-CONV | Resistance associated with within-canopy convection. | $r_{dc} = 100(1 + \dfrac{1000}{Rad})$ |




| Prescribed values (Table data) [pollutant, season] |
|---|
| $r_{cl}$: for exposed surfaces in the lower canopy $SO_2$, $O_3$ |
| $r_{ac}$: for transfer that depends on canopy height and density |
| $r_{gs}$: for ground surfaces $SO_2$, $O_3$ |
| $r_{si}$: for stomatal resistance |
| $r_{lu}$: for outer surfaces in the upper canopy |
| H: Henry´s law constant |
| $f_i$: Reactivity factor |






Example 7: CHIMERE

Figure B7. Resistance diagram for the dry deposition scheme implemented in CHIMERE

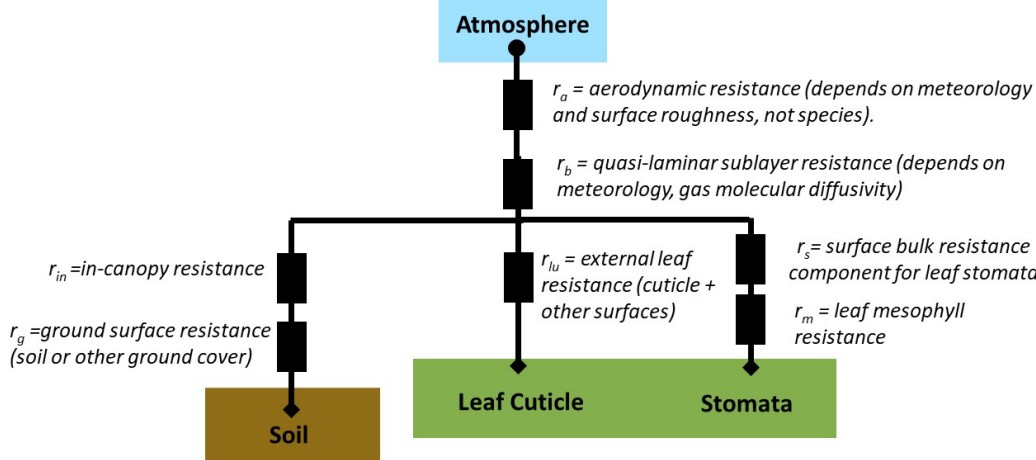






Table B7: AQMEII4 reported gaseous deposition variables corresponding to the CHIMERE resistance model of Figure
B7

| Name as described here | AQMEII4 Variable Name | Formulae |
|---|---|---|
| $r_a$ | RES_AERO | $RES\_AERO = r_a$ |
| $r_b$ | RES_QLSB | $RES\_QLSB = r_b$ |
| $r_c$ | RES_SURF | $RES\_SURF = \left( (r_s + r_m)^{-1} + (r_{lu})^{-1} + \left(r_{in} + r_g\right)^{-1} \right)^{-1}$ |
| $r_s$ | RES_STOM | $RES\_STOM = r_s$ |
| $r_m$ | RES_MESO | $RES\_MESO = r_m$ |
| $r_c$ | RES_CUT | $RES\_CUT = r_{lu}$ |
| $E_{STOM}$ | ECOND_ST | $ECOND\_ST = \left( \dfrac{(r_s + r_m)^{-1}}{(r_s + r_m)^{-1} + (r_{lu})^{-1} + \left(r_{in} + r_g\right)^{-1}} \right) V_d$ |
| $E_{CUT}$ | ECOND_CUT | $ECOND\_CUT = \left( \dfrac{(r_{lu})^{-1}}{(r_s + r_m)^{-1} + (r_{lu})^{-1} + \left(r_{in} + r_g\right)^{-1}} \right) V_d$ |
| $E_{SOIL}$ | ECONC_SOIL | $ECOND\_SOIL = \left( \dfrac{\left(r_{in} + r_g\right)^{-1}}{(r_s + r_m)^{-1} + (r_{lu})^{-1} + \left(r_{in} + r_g\right)^{-1}} \right) V_d$ |
| $E_{LCAN}$ | ECONC_LCAN | $ECOND\_LCAN$ $= -9\ (not\ included\ as\ a\ separate\ deposition\ pathway)$ |





**Appendix C. Bidirectional Ammonia Fluxes**

If a bidirectional flux algorithm for ammonia is employed in the model, then the flux may be either downwards
(defined positive here) or upwards (defined negative, here). The generic equation for the bidirectional flux with this
directionality is:
$$F_T = \frac{c_a - c_c}{r_{sum}} \qquad (7)$$
Where $F_T$ is the net flux, $c_a$ and $c_c$ are the atmospheric and canopy compensation point concentrations of ammonia
gas, and $r_{sum}$ is a sum of resistances. Different sources in the literature make use of different formula for both $c_c$ and
$r_{sum}$. For example, Zhang et al (2010) employs:
$$r_{sum} = r_a + r_b, and$$
$$c_c = \frac{\frac{c_a}{r_a+r_b}+\frac{c_s}{r_s}+\frac{c_g}{r_{ac}+r_{gs}}}{(r_a+r_b)^{-1}+(r_s)^{-1}+(r_{ac}+r_{gs})^{-1}+(r_{lu})^{-1}} \qquad (8)$$

Where $c_s$ and $c_g$ are compensation point concentrations relative to stomata and ground, respectively, and all other
terms are defined as above. CMAQ with the M3dry deposition option uses (Bash et al. 2013, Pleim et. 2013, Pleim
et al., 2019):
$$r_{sum} = r_a + 0.5\, r_{inc}$$
$$r_{inc} = 14 LAI \frac{h_{can}}{u_*} \; (based\ on\ Erisman, 1994) \qquad (9)$$
$$c_c = \frac{-B + (B^2 - 4AC)^{0.5}}{2A}$$

Where
$$A = r_{wet} G_t$$
$$B = r_{wb} G_t + LAI(1 - f_{wet}) - r_{wet}(G_a c_a + G_{sb} c_s + G_g c_g) \qquad (10)$$
$$C = -r_{wb}(G_a c_a + G_{sb} c_s + G_g c_g)$$

And
$$G_a = (r_a + 0.5 r_{inc})^{-1}$$
$$G_{sb} = (r_s + r_b)^{-1}$$
$$G_g = (r_{bg} + 0.5 r_{inc} + r_{soil})^{-1}$$
$$G_t = G_{sb} + G_g + G_a + f_{wet} G_{cw} \qquad (11)$$
$$G_{cw} = \frac{LAI}{r_b + r_{wet}}$$
$$r_{wet} = \frac{R_{wo}}{H_{eff}}$$
$$r_{wb} = r_{wet} + LAI[a_h(1 - f_{RH_s}) + r_b]$$

Where the terms $r_{soil}$, $H_{eff}$, $a_h$, $f_{RH_s}$, and $R_{wo}$ are defined in Pleim *et al.* (2013). Note that in the latter reference (their
equation (20)), the summation term in (10) above $G_a c_a$ is repeated twice within the bracketed terms (i.e.
$(G_a c_a + G_{sb} c_s + G_g c_g)$ as above is written $(G_a c_a + G_{sb} c_s + G_a c_a + G_g c_g)$ , but this second occurrence of $G_a c_a$ is
likely a typo).
CMAQ with the STAGE deposition option closely follows the widely used Massad et al. (2010) and Nemitz et al. (2001)
parameterizations modified to include the option for a cuticular compensation point and employs the same
resistance model for all deposited species as it reduced to RES-SURF from table B3 when the stomatal, $C_s$, cuticular,
$C_{cut}$, and ground, $C_g$, compensation points are zero. $NH_3$ bidirectional flux from the cuticle has been shown to be
important (cuticular $NH_3$ reference) however parameterizations applicable in a regional-scale model do not yet exist.





$$r_g = r_{dc} + r_{gnd,qlsb} + r_{gs} \tag{12}$$

$$r_{sum} = r_a \tag{13}$$

$$c_c = \frac{\frac{c_a}{r_a} + \frac{c_{leaf}}{r_{can,qlsb}} + \frac{c_g}{r_g}}{(r_a)^{-1} + (r_{can,qlsb})^{-1} + (r_{dc} + r_{gnd,qlsb} + r_{gs})^{-1}} \tag{14}$$


$C_{leaf}$ is the leaf compensation point and is estimated by solving for the exchange between the canopy compensation
point and the atmosphere, stomata, cuticle and ground following Kirchhoff's current law (e.g. Nemitz *et al.* 2000).
$C_{leaf}$ is solved from this system of equations as:
$$c_{leaf} = \frac{\frac{c_a}{r_a r_{can,qlsb}} + \frac{c_s}{r_a r_s + r_{can,qlsb} r_s + r_g r_s} + \frac{c_{cut}}{r_a r_{cut} + r_{can,qlsb} r_{cut} + r_g r_{cut}} + \frac{c_g}{r_{dc} + r_{gnd,qlsb} + r_{gs}}}{(r_a r_{can,qlsb})^{-1} + (r_a r_s)^{-1} + (r_a r_{cut})^{-1} + (r_{can,qlsb} r_s)^{-1} + (r_{can,qlsb} r_{cut})^{-1} + (r_{can,qlsb} r_g)^{-1} + (r_g r_s)^{-1} + (r_g r_{cut})^{-1}} \tag{15}$$


The resistances $r_{cut}$, $r_{can,qlsb}$, and $r_{gnd,qlsb}$ are taken from Massad et al. 2010, $r_{dc}$ follows Shuttleworth and Wallace (1985)
but integrated the canopy transport model of Yi 2008 using the in-canopy eddy diffusivity of Bash et al. 2010 from
the soil surface to top of the canopy and assuming $r_a = p_r U/u_*^2$ , the remainder of the resistances are the same as
CMAQ with the M3dry deposition option.
$$r_{dc} = r_a \left( e^{\frac{LAI}{2}} - 1 \right) \tag{16}$$


Comparing approaches (8 through 16), $r_{sum}$, $r_a$, and $c_c$ are held in common, and these approaches also make use of a
stomatal ($c_s$) and ground ($c_g$) compensation point concentration, although how these terms are combined varies
considerably between these approaches. For this reason, these common terms are reported as a separate TSD for
ammonia bidirectional fluxes in AQMEII4 in order to allow cross-comparison of different approaches.

Note that the net flux of ammonia $F_T$ appears as DFLUX-NH3 in the AQMEII4 documentation provided to participants
as TSDs and may be positive or negative depending on direction. Ammonia values for $r_b$, net canopy resistance,
stomatal resistance, mesophyll resistance, cuticle resistance and the three effective conductances also appear
elsewhere in the TSDs, both for the grid scale and by AQMEII4 LULC category.