# Peer review of "Technical Note – AQMEII4 Activity 1: Evaluation of Wet and Dry Deposition Schemes as an Integral Part of"

_Atmospheric Chemistry and Physics, 2021_

## Author Comment (AC1)

Reply to reviewer 1

We thank the reviewer for the careful reading and sensible suggestions to improve the manuscript. Follows are point-by-point replies to the comments. Replies are in italic font.

L203: It would be useful to have a figure to illustrate the European and North American domains.

*The figure has been added to the text*

L229-231: I do not understand what the authors mean by 'hourly speciated files'. Please clarify.

*This has been corrected, thank for pointing it out. The new section of text now reads:*

*"These EMP were used by the US EPA to generate 8 different hourly files of speciated emissions for each day in 2010 (1 gridded file with low-level emissions and files with elevated sources from 7 different sectors) and 9 different hourly speciated files for each day in 2016 (1 gridded file with low-level emissions and files with elevated sources from 8 different sectors) which were then shared with all participants."*

L320-330: Suggested text modification from (units s cm-1) to (units = s cm-1)

*Done.*

Figure 3: The resolution of this figure is generally poor and in particular the text for the x-axes tick labels is difficult to read (in fact unintelligible for 3a and 3d). This should be improved before final publication.

*This correction has been implemented.*

Can the authors explain why there is a smaller effective conductance for soil (and possibly lcan and cut, although it is difficult to tell from the figure) at 06:00?

Note that the former Figure 3 is now Figure 4. We have added the following paragraph of text to the document to address this point:

*"Also with reference to Figure 4, it should be noted that the effective conductances and effective fluxes show the relative contributions of the pathway towards the total deposition or the total flux at any given time, and that the net surface resistance appearing in the denominator of these terms may drive the time variation. For example, the soil effective conductance of Fig. 4 e minimizes at 6 AM – however, the factors contributing to the soil pathway itself for the model used in this example (see Appendix Table B1) are relatively time-invariant (seasonally varying). The temporal variation is driven by hourly variation in the stomatal term, and hence the relative importance of the soil conductance varies with time in Fig. 4 e. "*

Table 5: The format of this table should be improved. It might be better in landscape as all the columns could do with being a bit wider.

*The original table included as its second column the AQMEII4 variable name. We realized that name was repeated in the original table's 3rd column. We got rid of the original table's second column as being redundant, in order to clear up more space, and have re-titled the column with the formulae "AQMEII-4 Name = resistance diagram variable or formula". Ditto for Table 6.*

Tables 5 and 6: For ease of comparison, would it possible to situate Fig 2a in closer proximity to Table 5?

*The table and figure organization rest with the ACP's copyediting and typesetting department, We are sure they will accommodate the most readable solution*

L524-526: 'In this example, note that the branch containing the rdc term has been designated as the lower canopy pathway, due to the presence of the canopy buoyant convection term rdc (i.e., closest analogy to Wesely's setup is to have the pathway involving deposition to "soil" pathway is designated as a "lower canopy" pathway).'

=> Consider re-wording this sentence from 'the branch containing the rdc term' to 'the branch representing deposition to soil' or similar to avoid confusion about the two slightly different usages of rdc in this sentence.

*Changed to: "In this example, note that the branch representing deposition to soil is designated as the lower canopy pathway, due to the presence of the canopy buoyant convection term rdc (i.e., closest analogy to Wesely's setup is to have the pathway involving deposition to "soil" pathway designated as a "lower canopy pathway").*

Table A1: Should the units for the water vapour column be changed from cm3 cm-2 to cm3 cm-2? Units for RHO (Air density of lowest model layer)?

*We apologies but we do not really understand the request as the units proposed are the same as those adopted. One possibility we noted was that the initial "3" in "cm3" lacked the superscript (it's possible that the reviewer's original version had this change and was lost in copying to ACPD's web portal) – this has been corrected in the final version.*

Table A2: Change 'Number concentration of PM2.5 at ground, cm-3' to 'Number concentration of PM2.5 at ground, cm-3'. Units of eq ha -1?

*The issue has been corrected.*

Table A2: Could the authors please provide a description of the units eq ha -1.

*Also this has been fixed, thanks for pointing it out. New text has been added which reads, "Note that the units of nitrogen and sulphur deposition in Table A3 are "equivalents" per hectare per year, where the "equivalent" refers to the product of moles and the oxidized charge associated with the deposited species. All species depositing sulphur are assumed to have a charge of 2, all species depositing nitrogen to have a charge of 1. These units are used in the calculation of exceedances of critical loads, where the annual charge balance and flux of charge to ecosystems is used to estimate potential ecosystem impacts. "*

Appendix tables B1 – B7: I think the formatting could be improved across these tables and importantly, made consistent. E.g. Font formatting, equation layout (for preference, the equations should be as in Table B6, represented as fractions rather than as (xxxx)-1, but consistency is the main thing). In some cases the tables may be better displayed in landscape so that equations can be presented on one line. Table B6 is nicely laid out, although the text in column one is too small to read easily.

*The tables have been corrected as per the reviewer's suggestion to use fractions rather than (xxxx)-1, though we note again the publishing office may also modifiy the equations according to their internal policies. Thank you for your concern*

Table B2 and B3: Should RES – SURF be RES-SURF?

*Corrected thanks*

Figure B3 and B4: Shold these be the same?

*Thanks for picking up on this! We carefully went over all Figures and Tables in the Appendix again. We consulted with our LOTOS-EUROS co-author, and have corrected Figure B4 and Table B4 to match the LOTOS-EUROS model. We have also revised Table B3 – some of the formulae needed to be corrected, based on our check and discussion with participant co-authors.*

Appendix C, Equations 8 and 9: These are hard to decipher, please consider improving their layout

*The equations in Appendix C and some of the associated description have been revised to improve clarity of the description. We have revised the text to make the changes more clear with regards to the three bidirectional NH3 flux algorithms presented, along with revising the equation format to match that of the tables; using fractions more clearly. All of the formula in the Tables have been revised to be represented as fractions*

L958: Fix reference formatting for Yi (2008) and Bash et al., (2010)

Corrected

---

## Author Comment (AC3)

Reply to reviewer 2

We thank the reviewer for the careful reading and sensible suggestions to improve the manuscript. Follows are point-by-point reply to the comments. Replies are in italic font.

Versions and references for the modeling systems should be identified in Table 1.

*References have been added*

Maps of the European, North American, and combined domains should be added.

*Map added as new Figure 1 (all other Figure numbers incremented by 1).*

Please explain what is meant in lines 194-196 on page 7 regarding "past use in policy-relevant emissions scenario simulation, with changes in emissions policies that may affect the deposition".

*The has now been changed into : "The NA years were selected due their policy-relevance; the years 2010 and 2016 have featured in policy-relevant emissions scenario simulations by governments in the continent." Hopefully this clarifies the intent of the sentence.*

It would be helpful to know which emissions models were used, not only the source of the data, i.e., 2011v6.3 and 2016 beta, in the unified approach for forest fire emissions in North America and Canada.

*We've revised the section 2.2.1 slightly to mention the sources of information in more detail, with references for the biomass burning emissions data. The overarching 2010 and 2016 emission modeling platform URLs shown in section 2.2.1 (https://www.epa.gov/air-emissions-modeling/2011-version-63-platform, https://www.epa.gov/air-emissions-modeling/2016v72-beta-and-regional-haze-platform, and http://views.cira.colostate.edu/wiki/wiki/10197) provide more detailed information on the emissions inventory and processing approaches taken to generate these platforms. This includes information on the generation and processing of wildfire emissions with the exception of Canada 2010. Specifically, SMARTFIRE2 was used for U.S. wildfire emissions in both 2010 (https://www.epa.gov/sites/default/files/2016-09/documents/2011v6_3_2017_emismod_tsd_aug2016_final.pdf) and 2016 (http://views.cira.colostate.edu/wiki/Attachments/Inventory%20Collaborative/Documentation/2016beta_0311/National-Emissions-Collaborative_2016beta_point-fire_11Mar2019.pdf), while a combination of FireWork and FINN was used for Canada in 2016 (http://views.cira.colostate.edu/wiki/Attachments/Inventory%20Collaborative/Documentation/2016beta_0311/National-Emissions-Collaborative_2016beta_canada-mexico-ptfire_07Mar2019.pdf). The generation of the 2010 wildfire emissions for Canada is described in Chen et al. (2013) and this reference has been added to the revised manuscript.*

In lines 338-346 on page 13, consider summarizing and briefly explaining some of the key motivating factors that have led to the development of different resistance frameworks. For example, is it the evolution of measurement systems, availability of observational data, inclusion of missing deposition pathways, etc.

*This portion of text has been added:*

*"Several motivating factors likely led to the development of a diversity of resistance frameworks. In the intervening years subsequent to Wesely's introduction of the resistance framework concept, new measurement capabilities (for higher time resolution information, for greater chemical speciation, higher precision measurements) allowed the original algorithms to be tested and modified. Developments in plant physiology understanding have also resulted in improved stomatal resistance parameterizations. Examples include the observation-based introduction of bidirectional fluxes for ammonia gas, and improved understanding of the role of $CO_2$ fluxes in the deposition of other gases. Also, some divergence in approaches is likely due to algorithm developments having been made in the context of specific regional models — each of which encompasses a diverse range of process representation algorithms, vertical resolutions, horizontal resolutions, etc.. An algorithm which provided good performance relative to surface concentration observations within the context of one regional model thus may not have resulted in as good performance in another model, further spurring model-specific development. These factors have resulted in the variety of approaches for gas-phase deposition in current regional models, and provide the part of the motivation for this first attempt at cross-comparing the results of the models' deposition algorithms in detail — to show and explain the causes for these differences."*

Please describe the characteristics that are used to guide the mapping to each generic land use/land cover category. For example, what are the definitions of mixed forest or herbaceous cover?

*We thank the reviewer for the request for clarification - there is a whole discipline behind this issue that has its pinnacle research in remote sensing using satellite data. In order to carry out a meaningful comparison on a land-use category basis, common land-use categories had to be devised. However, the assignment of land use categories from the native model categories to the AQMEII4 set is of necessity a source of uncertainty. The need for cross-comparison of the land use category assignments is one of the reasons why both the assigned AQMEII-4 land use categories and the original model land use categories are included in the reported information from the participants. The additional text is as follows:*

*"We also note that the mapping of LULC types from the individual model land use classifications to the AQMEII-4 land use classifications is an unavoidable source of uncertainty in the land-use specific diagnostics. The 15 AQMEII-4 land use types themselves were based on a survey of land-use classifications used in 17 regional models. For example, while "Herbaceous" is available as an AQMEII-4 land use category, its intent is for use for moors and heathlands, while AQMEII-4 land use category "Wetlands" encompasses wetlands which are diversely described in individual model land use categories as herbaceous, wooded, and permanent wetlands, as well as swamps, and peatbogs. However, some categories were held in common by most models (e.g. Evergreen needleleaf forest, Deciduous broadleaf forest, snow and ice, mixed forest (usually taken as a combination of needleleaf and deciduous forests)), while others could easily be classified according to the broader landscape type of which they were a member (e.g. different types of Tundra were recommended to be classified as the AQMEII-4 Tundra*

*classification). Both the AQMEII-4 and "native model" land use types were reported by participants – with the aim using both sets of information to determine the extent to which land use database variation may be a factor in estimating deposition velocities, and to provide information on specific land use types used by specific models when these differences appear to be large."*

Check the references as well as the definitions of variables in Appendix B (which should be at the first instance) for completeness.

*Done. All Tables and Figures in the Appendix were reviewed for consistency. A number of corrections were made to the formulae in the Tables for the CMAQ-STAGE and LOTOS-EUROS model, and the description of Zhang's model has also been corrected. The CHIMERE model has been removed from the list of models (original submission Figure B7 and Table B7) since that participant stated that they will be unable to submit model results to AQMEII-4 in the intervening time since the Technical Note was submitted. Our other reviewer requested that we use fractions rather than (xxxx)-1 notation to make the formulae more clear and we've followed through with this. Another round of updates to these and the references may also result from ACP's copyediting and typesetting processes, to improve clarity.*